# The spectral form factor in the 't Hooft limit – Intermediacy versus universality

Ward L. Vleeshouwers[1,2]* and Vladimir Gritsev[1,3]

**1** Institute for Theoretical Physics, Universiteit van Amsterdam, Science Park 904, Postbus 94485, 1098 XH Amsterdam, The Netherlands
**2** Institute for Theoretical Physics, Universiteit Utrecht, Princetonplein 5, Postbus 80.089, 3584 CC Utrecht, The Netherlands
**3** Russian Quantum Center, Skolkovo, Moscow, Russia

* w.l.vleeshouwers@uva.nl

## Abstract

The Spectral Form Factor (SFF) is a convenient tool for the characterization of eigenvalue statistics of systems with discrete spectra, and thus serves as a proxy for quantum chaoticity. This work presents an analytical calculation of the SFF of the Chern-Simons Matrix Model (CSMM), which was first introduced to describe the intermediate level statistics of disordered electrons at the mobility edge [1]. The CSMM is characterized by a parameter $0 \leqslant q \leqslant 1$, where the Circular Unitary Ensemble (CUE) is recovered for $q \to 0$. The CSMM was later found as a matrix model description of $U(N)$ Chern-Simons theory on $S^3$ [2], which is dual to a topological string theory characterized by string coupling $g_s = -\log q$. The spectral form factor is proportional to a colored HOMFLY invariant of a $(2n, 2)$-torus link with its two components carrying the fundamental and antifundamental representations, respectively. We check explicitly that taking $N \to \infty$ whilst keeping $q < 1$ reduces the connected SFF to an exact linear ramp of unit slope, thereby confirming the main result from [3] for the specific case of the CSMM. We then consider the 't Hooft limit, where $N \to \infty$ and $q \to 1^-$ such that $y = q^N$ remains finite. As we take $q \to 1^-$, this constitutes the opposite extreme of the CUE limit. In the 't Hooft limit, the connected SFF turns into a remarkable sequence of polynomials which, as far as the authors are aware, have not appeared in the literature thus far. A gap opens in the spectrum and, after unfolding by a constant rescaling, the connected SFF approximates a linear ramp of unit slope for all $y$ except $y \approx 1$, where the connected SFF goes to zero. We thus find that, although the CSMM was introduced to describe intermediate statistics and the 't Hooft limit is the opposite limit of the CUE, we still recover Wigner-Dyson universality for all $y$ except $y \approx 1$.

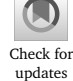
# 1 Introduction

Random matrix theory provides a phenomenological description of a wide variety of systems appearing in physics and beyond, starting with nuclear physics [4], and later finding applications in quantum chaos [5], disordered electronic [6] and mesoscopic [7] systems, chromodynamics [8], models of 2d quantum gravity and string theory [9], economics [10], information theory [11], and number theory [12]. One of the central concepts in the theory of random matrices is the Wigner-Dyson (WD) universality that is exhibited by the eigenvalue correlation properties of a wide variety of ensembles. For ensembles defined by the probability density with the weight function $w(x) = \exp(-V(x))$ where $V(H) \sim \mathrm{tr}(H^2) + \ldots$ for Hermitian $N \times N$ matrix $H$, the density of eigenvalues $x$ has the famous, "universal" semicircular form $\rho(x) = \sqrt{2N - x^2}/\pi$. This does not depend on the precise form of $V(x)$ for a broad range of potentials and the matrix size $N$. Rescaling the eigenvalues such that $\bar{\rho}(u) = 1$ near the origin, typically called unfolding, leads to the two-level eigenvalue kernel function $G(u, v) = \sin[\pi(u - v)]/\pi(u - v)$. As a consequence, this leads to the universal form of the Spectral Form Factor (SFF), a Fourier transform of the density-density correlation function, $K(\xi) = 1 - \int G(x, 0)^2 \exp(ix\xi)dx$ resulting for $\xi > 0$ in the linear ramp until the Heisenberg "time" $\xi = T_H = 2\pi$ followed by a plateau for $\xi > T_H$. Integrable systems, on the other hand, display uncorrelated Poisson statistics leading to a constant SFF for all times. For systems with statistics intermediate between Poissonian and WD, the SFF has mixed features. Typically, it is of a dip-ramp-plateau shape, where, for small $\xi$, the SFF dips until it reaches a minimum at Thouless time, after which it transitions into a linear ramp which saturates at a plateau at Heisenberg time. These features make the SFF a convenient tool for characterizing level statistics which has found extensive use in the RMT and quantum chaos community. We also mention here that the SFF has recently became popular in the string theory community, predominantly in the context of the AdS/CFT correspondence, as well as the SYK model and JT-gravity [13–18].

Certain systems which are somewhere in between chaotic and integrable, such as disordered electrons at the mobility edge of Anderson localization [19], or pseudo-integrable billiards [20], display so-called intermediate statistics. Various random matrix ensembles have been introduced which exhibit these statistics. To the best of our knowledge, the first such model was a solvable random matrix ensemble interpolating between Poisson and Wigner-

Dyson statistics that was introduced by Gaudin [21]. The same model was rediscovered later by Yukawa [22] in the context of a so-called Pechukas-Yukawa gas. Another type of RME that was introduced to describe intermediate statistics are the so-called banded RME's [23], for which the entries of the random matrices decay in a power-law fashion away from the main diagonal. Another such is the Moshe-Neuberger-Shapiro (MNS) ensemble [24], where the unitary invariance of the ensemble is explicitly broken by a potential term involving a fixed matrix, typically chosen to be diagonal. Bogomolny et al. [25] discovered a number of ensembles which are based on the Lax matrices of various integrable systems. These ensembles exhibit intermediate statistics and multifractal eigenfunctions [26]. Further, a generalization [27] of the Rosenzweig-Porter model [28] was shown to have a region in parameter space with multifractal behavior. Moreover, a standard $\beta$-ensembles, introduced in [29] have intermediate statistics [30].

Another class of matrix models was proposed in [1,31] to describe intermediate level statistics. These ensembles are defined in terms of a $U(N)$-invariant measure, as opposed to the aforementioned ensembles with intermediate statistics. It was observed that WD-universality is lost for sufficiently shallow confining potentials, which asymptotically behave as $V(H) \sim g_s^{-1} \log^2 H$ for $|H| \gg 1$. Such potentials are associated with indeterminate moment problems, which is to say that the weight function $w(x)$ is not uniquely determined by its moments $m_j = \int x^j w(x) dx$ [32,33]. A typical signature of this intermediate class of statistics is a lack of a simple translationally-invariant kernel of $\sin x/x$-form as before, which now becomes dependent on some parameter $q$. This opens up the possibility of non-unique (and more complicated) scaling limits, [34,35].

The connection of the CSMM with intermediate statistics was seen to arise in earlier works as follows. It was shown that taking $N \to \infty$ and then $q = e^{-g_s} \to 1$ leads to a kernel that is of the form $G(\theta, \varphi) = \frac{g_s}{2\pi} \frac{\sin[\pi(\theta-\varphi)]}{\sinh[g_s(\theta-\varphi)/2]}$ [1,32]. This was found to be precisely the kernel [31] for the aforementioned banded random matrix ensembles and the MNS-ensemble. The relation of the ($U(N)$-invariant) CSMM to the (non-$U(N)$-invariant) banded and MNS-ensembles has been argued to arise due to a spontaneous breaking of $U(N)$-invariance [36]. It was found that the CSMM reproduces both the nearest neighbor spacing and the level number variance at the same value of the parameter $q$ [37]. From the level number variance, one can determine certain properties of the eigenvector statistics as well, in particular their multifractal dimension(s) [38,39]. The SFF was previously calculated using the limiting sinh-kernel in [40]. The CSMM appears in unitary guise as a one-parameter dependent generalization of the circular unitary ensemble (CUE), see e.g. [41] for a map between the unitary and Hermitian ensembles. Denoting the parameter as $q = e^{-g_s}$, the CSMM reduces to the CUE for $q \to 0$ and produces Poissonian statistics for $q \to 1$ such that $q^N = 1$. This paper focuses on the unitary version of the CSMM. The CUE also exhibits WD-universality, consisting of a linear ramp of unit slope that saturates at a plateau [42].

The CSMM was later found by Mariño as a matrix model representation of type $A$ topological open string theory on the cotangent space of $S^3$ [2], which reduces to $U(N)$ Chern-Simons theory on $S^3$ [43]. The orthogonal polynomials associated to this ensemble are the Stieltjes-Wigert or Rogers-Szegö polynomials in Hermitian and unitary description, respectively [33,44]. Witten famously showed that Wilson line expectation values in $SU(2)$ Chern-Simons theory equal topological (Jones) invariants of knots and links [45], which was later generalized to $U(N)$ for general $N$. Indeed, the SFF, $\langle |\text{tr} U^n|^2 \rangle$, is proportional to the HOMFLY invariant of a $(2n, 2)$-torus link with one Wilson line in the fundamental and the other in the antifundamental representation.

In the string theory literature on the CSMM, the 't Hooft limit has been considered, where one takes $N \to \infty$ and simultaneously $q \to 1$ such that $y = q^N$ remains finite. This idea goes back to pioneering work by 't Hooft in the context of $U(N)$ gauge theories at large $N$. In the

type $A$ topological open string theory on $T^*S^3$ described by the $U(N)$ Chern-Simons theory, certain large $N$ dualities appear. In particular, it has been argued that the topological $A$-type open string theory on $T^*S^3$ undergoes a conifold transition to a *closed* type $A$ topological string theory on the resolved conifold [46]. The magnitude of the $B$-field on the $S^2$ blowup of the conifold is given by $t = Ng_s$, the 't Hooft parameter. The mirror dual of the conifold geometry can be seen to arise from the resolvent of the matrix model in the large $N$ limit [47,48], see also [49,50]. These dualities and related results have important applications in enumerative geometry and intersection theory.

As far as the authors are aware, the 't Hooft limit has heretofore not been explicitly considered for the CSMM in the RMT literature. In [51], a closely related limit was considered for a Hermitian version of the $q$-deformed ensemble considered here, where the weak disorder (GUE) limit corresponds to $q = e^{-\gamma} \to 1$ and $N \to \infty$ such that $\gamma N \to 0$, while the strong disorder limit involves $\gamma N = $ constant. In the latter limit, which is essentially the 't Hooft limit, an approximate expression for the parametric density correlation function was found in [51]. Further, a similar limit was considered for another, closely related, $q$-deformed circular unitary ensemble in [52], see also [32]. It was found that deviations from the CUE level density only persist in the infinite $N$ limit if one simultaneously scales $q$ such that $(1-q)N$ remains finite, which is essentially the 't Hooft limit.

In [3], we calculated the spectral form factor (SFF) for $N \to \infty$ invariant unitary matrix models satisfying the assumptions of Szegö's theorem. In this limit, we found that for *all* such ensembles, the SFF is of a surprisingly simple form, where the connected SFF always consists of an exact linear ramp and plateau, and the disconnected part constitutes a dip which consists of a squared simple power sum polynomial with of variables that can be read off from the weight function. In the present work, we extend the calculation of the SFF for the CSMM to finite $N$ and explore first how our previous results are recovered in the $N \to \infty$. The present calculation proceeds from the expansion of the SFF in terms of Toeplitz minors, which are proportional to the product of two hook-shaped Schur polynomials of different specialization. The simplification that occurs for $N \to \infty$ [53] is not present here, which precludes us from generalizing these results to other matrix models as we did for infinite $N$ [3].

We then proceed to take the 't Hooft limit, where $N \to \infty$ and $q \to 1$. Since the CSMM was introduced to describe intermediate statistics and reduces to the CUE for $q \to 0$, one would naturally expect the CSMM to exhibit deviation from WD-universality in the opposite limit, i.e. $q \to 1$. This is our main physical motivation for studying the 't Hooft limit. We find that the connected SFF, which we denote by $F(n)_c$, turns into a remarkable sequence of polynomials of degree $2n - 1$ in $y = q^N$, which we computed for $n = 1, \dots, 11$. These polynomials do not seem to have appeared in the literature thus far. Their somewhat complicated form belies the fact that they are very close to linear ramps, with slope decreasing from 1 to 0 as we increase $y$ from 0 to 1. In fact, we find that the $F(n)_c$ is an exact linear ramp of slope $1/2$ for $y = 1/2$, with the SFF's for $y$ and $1 - y$ exactly adding up to a linear ramp of unit slope. Although the form of low-lying expansion coefficients can be found, the general structure of these polynomials (beyond those which calculated explicitly) remains elusive. Further, a gap opens in the eigenphase density, which we will simply refer to as level density, for any $y > 0$. We unfold by rescaling the spectrum so that the level density has support on an interval of length $2\pi$. After calculating the connected SFF using the unfolded eigenphases, we find that it is very close to a linear ramp of unit slope for all $y$ other than $y$ close to 1, with precision increasing with $n$. Indeed, we recover an exact linear ramp of *unit* slope for $y = 1/2$, in spite of the fact that our unfolding procedure involves only a rescaling, and the unfolded level density is not flat at all (in fact it closely resembles a semicircle). We thus recover WD-universality for $y$ sufficiently far from 1, in spite of the fact that the 't Hooft limit involves $q \to 1$, which is the opposite limit of the CUE limit. This the main result of the present work. Lastly, we

consider the non-commutativity of the limits $q \to 1$ and $N \to \infty$, which was already noted for the partition function and trace averages $\langle \mathrm{tr} U^n \rangle$ in [54].

The outline of the paper is as follows. In section 2, we review the calculation of $U(N)$ integrals "twisted" by the insertion of $U(N)$ characters (Schur polynomials), and we explicitly consider a few examples relevant for the calculation of the SFF. In section 3, we present the calculation of the SFF. In particular, in section 3.1, we calculate the SFF for general $N$ and $q$, and show how the connected SFF reduces to a linear ramp for $y = q^N \to 0$, thereby confirming one of the main results of [3]. We further demonstrate explicitly that the linear ramp emerges from $U(N)$ averages over characters corresponding to identical (hook-shaped) representations, which confirms a result derived in [3] for $N \to \infty$ with $q < 1$. In section 3.2, we demonstrate how a plateau emerges sufficiently far from the origin and explain its arising from the properties of Schur polynomials $s_\lambda$, in particular from the simple fact that $s_\lambda(x_1, \ldots, x_N) = 0$ if the number of non-empty rows in partition $\lambda$ exceeds the number of variables $N$. In section 3.3, we consider the 't Hooft limit and compute the SFF and level density. We unfold by a linear rescaling and demonstrate how WD-universality is recovered as a result. Then, in section 3.4, we explore the non-commutativity of the limits $N \to \infty$ and $q \to 1$ and calculate the SFF for small 't Hooft parameter. We finish this work by presenting our outlook and conclusions.

## 2 Twisted $U(N)$ integrals

Consider the weight function of some ensemble, expressed here as

$$f(z) \quad = \sum_{k \in \mathbb{Z}} d_k z^k = \prod_{j=1}^{\infty} (1 + x_j z)(1 + x_j z^{-1}) = E(x; z) E(x; z^{-1}), \tag{1}$$

where $x = (x_1, x_2, \ldots)$ are the variables of $E(x; z)$, the generating function of elementary symmetric polynomials. For $U \in U(N)$ with eigenvalues $e^{i\phi_j}$, we write

$$\tilde{f}(U) = \prod_{j=1}^{N} f(e^{i\phi_j}), \quad s_\lambda(U) = s_\lambda(e^{i\phi_j}). \tag{2}$$

In the limit $N \to \infty$, the twisted $U(N)$ integral goes to [55]

$$\langle s_\lambda(U^{-1}) s_\mu(U) \rangle := \frac{\int_{U(N)} \tilde{f}(U) s_\lambda(U^{-1}) s_\mu(U) dU}{\int_{U(N)} \tilde{f}(U) dU} = \sum_\nu s_{(\lambda/\nu)^t}(x) s_{(\mu/\nu)^t}(x), \tag{3}$$

where the superscript $^t$ denotes transposition and where the sum is over all partitions $\nu$ such that $\mu \supseteq \nu \subseteq \lambda$. Further, $x = (x_1, x_2, \ldots)$, the set of variables appearing in (1). If we have a finite number of distinct non-zero $x_j$, (3) can be valid even for finite $N$ [3]. We denote by $|x|$ the number of distinct, non-zero $x_j$. Then, (3) remains valid as long as both $\ell(\lambda)$ and $\ell(\mu)$ are greater than $N - |x|$. However, we are most interested in the CSMM, for which $f(z) = \Theta_3(z)$. Using the Jacobi triple product expansion, we see that $x_j = q^{j-1/2}$ in equation (1) for all $j \in \mathbb{Z}^+$, so that we cannot apply (3). In particular, the partition function of the CSMM can be written as

$$Z = \int dU \prod_{j=1}^{\infty} \det \left( \mathbb{1} + q^{j-1/2} U \right) \det \left( \mathbb{1} + q^{j-1/2} U^{-1} \right). \tag{4}$$

All expectation values $\langle \ldots \rangle$ that we write from now on are with respect to the above partition function. To calculate the SFF of the CSMM, then, we will consider twisted $U(N)$ integrals for

general $N$ and weight function. These can be expressed as a minors of a Toeplitz matrix of symbol $f(z) = \sum_{k \in \mathbb{Z}} d_k z^k$, i.e. a Toeplitz matrix with $d_k$ on the $k^{\text{th}}$ diagonal [56],

$$D_{N-1}^{\lambda,\mu}(f) = \det(d_{\lambda_j - j - \mu_k + k})_{j,k=1}^N = \int_{U(N)} \tilde{f}(U) s_\lambda(U^{-1}) s_\mu(U) dU. \tag{5}$$

The weight function under consideration in the present work is the third theta function, which, for $0 < |q| < 1$, can be expressed in the triple product expansion as

$$\begin{aligned}
\Theta_3(z) &= \sum_{k \in \mathbb{Z}} q^{k^2/2} z^k = (q;q)_\infty \prod_{j=1}^\infty (1 + q^{j-1/2} z)(1 + q^{j-1/2} z^{-1}) \\
&= (q;q)_\infty E(q^{j-1/2}; z) E(q^{j-1/2}; z^{-1}). 
\end{aligned} \tag{6}$$

Note that, with this definition, $d_k = q^{k^2/2}$ rather than $d_k = q^{k^2}$, the latter being another common convention. We then have

$$D_{N-1}^{\lambda,\mu}(\Theta_3) = \det\left(q^{(\lambda_j - j - \mu_k + k)^2/2}\right)_{j,k=1}^N. \tag{7}$$

First taking $\lambda = \varnothing = \mu$, we have

$$D_{N-1}(\Theta_3) = \det\left(q^{(k-j)^2/2}\right)_{j,k=1}^N = \prod_{j<k}(1 - q^{k-j}) = \prod_{k=1}^{N-1}(1 - q^j)^{N-j}. \tag{8}$$

Taking only $\lambda = \varnothing$ gives

$$D_{N-1}^\mu(\Theta_3) = \det(d_{k-\mu_k-j})_{j,k=1}^N = \int_{U(N)} \tilde{f}(U) s_\mu(U) dU = \det\left(q^{(k-\mu_k-j)^2/2}\right)_{j,k=1}^N. \tag{9}$$

We have (see e.g. the appendix of [54]),

$$\det(q^{(\mu_k + k - j)^2/2})_{j,k=1}^N = q^{\sum_k \mu_k^2/2} \prod_{j>k}(1 - q^{\mu_j - \mu_k + k - j}). \tag{10}$$

Then,

$$W_{\lambda\mu} := \left\langle s_\lambda(U^{-1}) s_\mu(U) \right\rangle = \frac{D_{N-1}^{\lambda\mu}}{D_{N-1}}. \tag{11}$$

Up to a simple framing factor, this equals the Chern-Simons average over a pair of Wilson lines tied into a Hopf link, where one Wilson line carries a $U(N)$ representation $\lambda$ and the other carries $\mu$. For $\mu = \varnothing$, the Hopf link reduces to an unknot carrying rep $\lambda$, and vice versa for $\lambda = \varnothing$. Using (10), we then have

$$\begin{aligned}
W_\mu = \frac{\det\left(q^{(k-\mu_k-j)^2/2}\right)}{\det\left(q^{(k-j)^2/2}\right)} &= q^{\sum_j \mu_j^2/2} \frac{\prod_{j<k}(1 - q^{k-j-\mu_k+\mu_j})}{\prod_{j<k}(1 - q^{k-j})} \tag{12} \\
&= q^{-n(\mu) + \sum_j \mu_j^2/2} s_\mu(1, q, \dots, q^{N-1}), \tag{13}
\end{aligned}$$

where $n(\mu) = \sum_{j=1}^N (j-1)\mu_j$. When we take $N \to \infty$, this equals [53]

$$W_\mu^\infty = s_{\mu^t}(q^{j-1/2}) = q^{|\mu|/2 + n(\mu^t) - n(\mu)} s_\mu(q^{j-1}). \tag{14}$$

The power appearing in the prefactor is given by the sum of the content, $c(x)$, over the partition $\mu$. Specifically, $c(x) = j - i$ for $x = (i, j) \in \mu$, and [57]

$$\sum_{x \in \mu} c(x) = n(\mu^t) - n(\mu). \tag{15}$$

For general $\lambda$ and $\mu$, finite $N$, and $|q| < 1$, we have

$$\begin{aligned}
W_{\lambda\mu} &= \frac{1}{Z_N} \int s_\lambda(U^{-1}) s_\mu(U) f(U) dU \\
&= q^{\sum_{j=1}^N (\lambda_j^2/2 + \mu_j^2/2 - (j-1)(\lambda_j + \mu_j))} s_\mu(q^{j-1}) s_\lambda(q^{-\mu_1}, q^{1-\mu_2}, \dots, q^{N-1-\mu_N}) \\
&= q^{-n(\lambda) - n(\mu) + \sum_{j=1}^N (\lambda_j^2/2 + \mu_j^2/2)} s_\mu(q^{j-1}) s_\lambda(q^{-\mu_1}, q^{1-\mu_2}, \dots, q^{N-1-\mu_N}),
\end{aligned} \tag{16}$$

which, for $N \to \infty$, goes to [53]

$$W_{\lambda\mu}^\infty = \sum_\nu s_{(\lambda/\nu)^t}(q^{j-1/2}) s_{(\mu/\nu)^t}(q^{j-1/2}). \tag{17}$$

Consider, for example, $\lambda = \square = \mu$. Then, using [equation (5.9) in [57]]

$$s_\lambda(x, y) = \sum_\mu s_{\lambda/\mu}(x) s_\mu(y), \tag{18}$$

and $\lim_{N \to \infty} [N]_q = \lim_{N \to \infty} \frac{1-q^N}{1-q} = \frac{1}{1-q}$ for $|q| < 1$, we have

$$W_{\square\square} = [N] + q^2 [N][N-1] \overset{N \to \infty}{\to} \frac{1}{1-q} + \frac{q^2}{(1-q)^2}. \tag{19}$$

On the other hand,

$$\begin{aligned}
W_{\square\square}^\infty &= \sum_\nu (s_{(\square/\nu)}(q^{j-1/2}))^2 = (s_\square(q^{j-1/2}))^2 + (s_\varnothing(q^{j-1/2}))^2 \tag{20} \\
&= q[N]^2 + 1 = \frac{q}{(1-q)^2} + 1 = \lim_{N \to \infty} W_{\square\square}. \tag{21}
\end{aligned}$$

As one can see, the fact that terms of the form $q^N$ go to zero as $N \to \infty$ leads to the agreement between these expressions. Consider the Schur polynomial appearing in the unknot and Hopf link, given by the $q$-hook length formula,

$$s_\lambda(q^{j-1}) = q^{n(\lambda)} \prod_{x \in \lambda} \frac{[N + c(x)]}{[h(x)]} = q^{n(\lambda)} \dim_q(\lambda), \tag{22}$$

where $h(x)$ is the hook-length of $x \in \lambda$. The quantity $\dim_q(\lambda)$ is known as the *quantum dimension*, or $q$-dimension. Its expression in (22) is simply the usual hook length formula where numbers are replaced by $q$-numbers $[N] = \frac{1-q^N}{1-q}$. Using the $q$-hook-length formula, one finds that the hook-shaped Schur polynomial is given by

$$s_{(a,1^b)}(x_i = q^{i-1}) = q^{b(b+1)/2} \frac{[N+a-1]!}{[N-b-1]![a-1]![b]![a+b]}. \tag{23}$$

For $c(x) \ll N$, $\forall x \in \lambda$ and $N \to \infty$, (22) gives

$$s_\lambda(q^{j-1}) = \frac{q^{n(\lambda)}}{(1-q)^{|\lambda|}} \prod_{x \in \lambda} [h(x)]^{-1} = q^{n(\lambda)} \prod_{x \in \lambda} (1 - q^{h(x)})^{-1}. \tag{24}$$

In particular, $s_\lambda(q^{j-1})$ depends only on $n(\lambda)$ and the hook lengths, so that e.g. $\frac{[N+a-1]}{[N-b-1]} = (1-q)^{-(a+b)}$. Therefore, for a partition $\lambda$ for which

$$\sum_{x\in\lambda} c(x) = n(\lambda^t) - n(\lambda) = 0,\tag{25}$$

the unknot $W_\lambda^\infty$ is invariant under taking $\lambda \to \lambda^t$. One can clearly see from equation (22) that this is not the case for finite $N$. These examples illustrate the simplification which occurs as $q^N \to 0$, which we will further comment on in the following section.

## 2.1 The 't Hooft limit

The 't Hooft limit is given by the following double scaling,

$$N \to \infty, \quad g_s \to 0, \quad \text{such that } t := Ng_s = \text{finite}.\tag{26}$$

The 't Hooft limit of the CSMM has been considered in the context of topological string theory in e.g. [46, 48, 49]. In this case, one has $y = q^N = e^{-t} \neq 0$. In this limit, $q$ taken to a finite power will simply give 1, whereas $q$ to the power of multiples of $N$ will give powers of $y$, which we need to keep track off to calculate the SFF. In the 't Hooft limit, the hook-shaped Schur polynomial in (23) goes to

$$\lim_{q\to 1^-} \frac{1}{(a-1)!b!(a+b)} \left(\frac{1-y}{1-q}\right)^{a+b},\tag{27}$$

which we write as a limit as it is a divergent quantity. However, we will find that the connected SFF is in fact not divergent for the explicit examples we calculated. For the connected SFF not to be divergent, a precise cancellation between various powers of $(1-q)^{-1}$ has to take place, which means that we cannot simply use (23) in this calculation. Instead, we will write,

$$s_{(a,1^b)}(x_i = q^{i-1}) = \underbrace{\frac{q^{b(b+1)/2}}{[a-1]![b]![a+b]}}_{=A_{a,b}} \frac{[N+a-1]!}{[N-b-1]!} = A_{a,b} \frac{\prod_{k=0}^{a+b-1}(1-yq^{a-1-k})}{(1-q)^{a+b}}!$$

$$= \frac{A_{a,b}}{(1-q)^{a+b}} (yq^{a-1}; q^{-1})_{a+b},\tag{28}$$

where one should keep in mind that we take the limit $q \to 1^-$. In particular, for $m$ finite, one can write

$$\begin{bmatrix} N+m \\ k \end{bmatrix} = \frac{(yq^m; q^{-1})_k}{[k]!(1-q)^k},\tag{29}$$

as a convenient way to extract factors of $y$.

# 3 Spectral form factor

We proceed to calculate the SFF, which is given by

$$K(n) := \frac{1}{N}\left\langle |\mathrm{tr}U^n|^2 \right\rangle = \frac{1}{N} \sum_{r,s=0}^{n-1} (-1)^{r+s} \left\langle s_{(n-r,1^r)} s_{(n-s,1^s)} \right\rangle,\tag{30}$$

where we applied the expansion of the power sum polynomial,

$$\mathrm{tr}U^n = \sum_{j=1}^{N} e^{in\phi_j} = \sum_{r=0}^{n-1} (-1)^r s_{(n-r,1^r)}(e^{i\phi_j}). \tag{31}$$

Plugging equation (30) into (16), we have

$$K(n) = \frac{q^{n^2}}{N} \sum_{r,s=0}^{n-1} (-1)^{r+s} q^{-n(r+s)} s_{(n-s,1^s)}(q^{j-1}) s_{(n-r,1^r)}(q^{-(n-s)},1,\ldots,q^{s-1},q^{s+1},\ldots,q^{N-1}). \tag{32}$$

The first Schur polynomial appearing in the sum, $s_{(n-s,1^s)}(q^{j-1}) = s_{(n-s,1^s)}(1,\ldots,q^{N-1})$, is given in (23). The second Schur polynomial on the right hand side of (32), of the form $s_\lambda(q^{j-\mu_j-1})$ for hook-shaped $\lambda$ and $\mu$, is more complicated. We write $\lambda = (a,1^b)$ and $\mu = (c,1^d)$. Defining the sets of variables $x = q^{-c}$, $y = q^{d+1},\ldots,q^{N-1}$, $z = 1,\ldots,q^{d-1}$, we use the following expression [equation (5.10) in [57]],

$$s_\lambda(x,y,z) = \sum_{\rho,\nu} s_{\lambda/\rho}(x) s_{\rho/\nu}(y) s_\nu(z), \tag{33}$$

where the sum runs over all partitions satisfying $\nu \subset \rho \subset \lambda$. For $\lambda = (a,1^b)$ and $x,y,z$ as defined above, we get non-zero contributions only when $\lambda/\rho$ is a horizontal strip, as $x = q^{-c}$ consists only of a single variable. We then have to carefully distinguish between two types of partitions $\rho$.

1. For $\rho = (e,1^b)$, we again have $\lambda/\rho = (a-e)$ which is obviously a horizontal strip. Then,

$$s_{\lambda/\rho}(q^{-c}) = q^{-c(a-e)}. \tag{34}$$

   The requirement that $\nu \subset \rho$ then gives $\nu = (f,1^g)$ so that $\rho/\nu = (e-f) \otimes (1^{b-g})$ for $\nu \neq \varnothing$ and $\rho/\nu = \mu$ for $\nu = \varnothing$.

2. For $\rho = (e,1^{b-1})$, we have $\lambda/\rho = \square \otimes (a-e)$ so that

$$s_{\lambda/\rho}(q^{-c}) = q^{-c(a-e+1)}. \tag{35}$$

   Then, $\nu = (f,1^g)$ so that $\rho/\nu = (e-f) \otimes (1^{b-g-1})$ and $\rho/\nu = \rho$ for $\nu = \varnothing$. This situation of course does not occur for $b = 0$, in which case $\lambda = (a)$.

Note that $s_{(a)}(x) = h_a(x)$ and $s_{(1^a)}(x) = e_a(x)$, the complete homogeneous and elementary symmetric polynomials of degree $a$, respectively. We then have, for $\rho/\nu = (e-f) \otimes (1^{b-g})$

$$s_{\rho/\nu}(y) = h_{e-f}(y) e_{b-g}(y). \tag{36}$$

We illustrate these two choices for $\rho$ in equations (34) and (35) for $\lambda = (4,1^2)$. As a specific example, we set $e = 2$, so that the first choice of $\rho = (e,1^b) = (2,1^2)$. This gives $\lambda/\rho = (4,1^2)/(2,1^2) = (2)$. $\lambda/\rho$ is represented in terms of Young diagrams below:

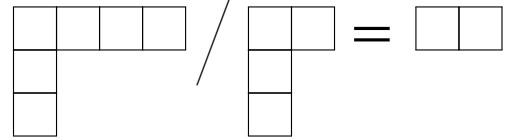

Consider now the second choice, $\rho = (e, 1^{b-1}) = (2,1)$, so that $\lambda/\rho = (4,1^2)/(2,1) = (2) \times (1)$, represented in Young diagrams as follows:

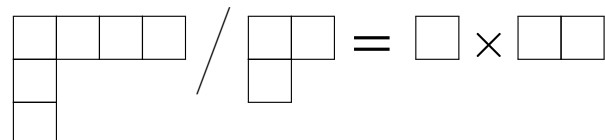

The result is again a horizontal strip, consisting in this case of two disconnected components. We consider these two choices for $\mu$ and sum over all partitions satisfying $\nu \subset \rho \subset \lambda$ to calculate $s_{(n-r,1^r)}(x,y,z)$[1] and, with that, the full SFF. We discuss the results of this calculation below.

## 3.1 Appearance of the linear ramp for $n < N$

For finite $N > n$, we consider how the linear ramp appears for the connected SFF, which is found by subtracting the disconnected contribution, $\langle \mathrm{tr} U^n \rangle^2$, from the full SFF. The disconnected contribution is thus given by the square of

$$\langle \mathrm{tr} U^n \rangle = q^{n^2/2} \sum_{r=0}^{n-1} (-1)^r q^{-nr} s_{(n-r,1^r)}(q^{j-1}). \tag{37}$$

For $n = 1, 2, 3$, this equals

$$\langle \mathrm{tr} U \rangle = q^{1/2}[N] = \frac{q^{1/2}(1-q^N)}{1-q},$$

$$\langle \mathrm{tr} U^2 \rangle = \frac{(1-q^N)(-q+q^N+q^{N+1}+q^{N+2})}{1-q^2}, \tag{38}$$

$$\langle \mathrm{tr} U^3 \rangle = \frac{q^3 - q^{N+1}(1+q+q^2)^2 + q^{2N}(1+q^2)(1+q+q^2)^2 - q^{3N}(1+q^2)(1+q+q^2+q^3+q^4)}{q^{3/2}(1-q^3)}.$$

We write,

$$F(n) = NK(n), \quad F(n)_c = NK(n)_c. \tag{39}$$

The connected SFF for small $n$ is then given by,

$$\begin{aligned} F(1)_c &= q\, s_{(1)}(q^{j-1}) s_{(1)}(q^{-1},q,q^2,\dots) - q(s_{(1)}(1,q,q^2\dots))^2 \\ &= q[N](1-q) = 1-q^N, \\ F(2)_c &= \frac{(1-q^N)(2q - q^N + q^{2N} - q^{2+N} + 2q^{1+2N} + q^{2+2N})}{q}, \\ F(3)_c &= -\frac{1}{q^4}(-3q^4 + q^{3N} - 2q^N + q^{5N} + q^{2+N} + 2q^{3+N} + 3q^{4+N} + 2q^{5+N} + q^{6+N} \\ &\quad - 2q^{1+2N} - 4q^{2+2N} - 8q^{3+2N} - 8q^{4+2N} - 8q^{5+2N} - 4q^{6+2N} - 2q^{7+2N} \\ &\quad + 6q^{1+3N} + 10q^{2+3N} + 16q^{3+3N} + 18q^{4+3N} + 16q^{5+3N} + 10q^{6+3N} + 6q^{7+3N} \\ &\quad + q^{8+3N} - 6q^{1+4N} - 12q^{2+4N} - 16q^{3+4N} - 18q^{4+4N} - 16q^{5+4N} - 12q^{6+4N} \\ &\quad - 6q^{7+4N} - 2q^{8+4N} + 2q^{1+5N} + 5q^{2+5N} + 6q^{3+5N} + 8q^{4+5N} + 6q^{5+5N} + 5q^{6+5N} \\ &\quad + 2q^{7+5N} + q^{8+5N}). \end{aligned} \tag{40}$$

---

[1]In particular, we sum over $g$ from 0 to $\min(b, d-1)$ or to $\min(b-1, d-1)$, corresponding to $\rho = (e, 1^b)$ or $\rho = (e, 1^{b-1})$, respectively. We then sum over $f$ from 1 to $e$ and lastly over $e$ from 0 to $a$.

Examples of the SFF for higher $n$ are too long to print here. One thing one can see from (40), which persists for higher $n$, is that the SFF is of the form,

$$F(n)_c = n + \mathcal{O}(q^A), \quad A = N + \dots \tag{41}$$

Therefore, for $N \to \infty$ and $q < 1$ fixed, $q^N \to 0$ and $F(n)_c \to n$. This reproduces the exact linear ramp that was found for all RME's satisfying the assumptions of Szegö's theorem in our previous work [3], thus recovering our main result for the case of the CSMM via a different computation.

### 3.1.1 Linear ramp from Schur bilinears

As mentioned above, as we take $N$ to infinity for $q < 1$, the connected SFF for $n/N < 1$ is a linear ramp of unit slope [3]. In this limit, in the expansion of the form factor in terms of averages of bilinears of hook-shaped Schur polynomials,

$$F(n) = \sum_{r,s=0}^{n-1} (-1)^{r+s} \left\langle s_{(n-r,1^r)}(U^{-1}) s_{(n-s,1^s)}(U) \right\rangle, \tag{42}$$

we get a contribution equal to 1 when considering two identical hook-shaped partitions $(n-r, 1^r) = (n-s, 1^s)$. Since there are $n$ hook-shaped partitions containing $n$ boxes, we get a contribution equal to $n$, which is the linear ramp. More details can be found in [3].

The above consideration leads us to conclude that, for finite $N$ and for $r = s \leqslant N - 1$, we should have that the summand of the SFF in (32) is of the following form

$$\begin{aligned}
A(N, n, q, r, r) &:= \left\langle s_{(n-r,1^r)} s_{(n-r,1^r)} \right\rangle \\
&= q^{n^2 - 2nr} s_{(n-r,1^r)}(q^{j-1}) s_{(n-r,1^r)}(q^{-(n-r)}, 1, \dots, q^{r-1}, q^{r+1}, \dots, q^{N-1}) \\
&= 1 + \mathcal{O}(q).
\end{aligned} \tag{43}$$

Further, we should have

$$A(N, n, q, r, s) = \mathcal{O}(q), \quad r \neq s. \tag{44}$$

There are two types of terms of $\mathcal{O}(q)$ in the above expressions. First of all, there are terms of the form $q^{N+\cdots}$, which go to zero as we take $N \to \infty$ for fixed $q < 1$. Secondly, there are powers of $q$ not containing factors of $N$, which do not go to zero as $N \to \infty$. Therefore, to recover the linear ramp as $N \to \infty$, all the lower powers of $q$ should mutually cancel out between the various terms in the sum in (32). It should be clear from equation (32) that the fact that such a cancellation occurs is a priori far from obvious.

We start by verifying (43). Note that $q$-numbers and products thereof (such as $q$-factorials and $q$-binomials) are themselves of $\mathcal{O}(1)$. For example,

$$[N]_q = \frac{1 - q^N}{1 - q} = (1 - q^N) \sum_{k=0}^{\infty} q^k = 1 + q + q^2 + \dots - q^N - q^{N+1} + \dots \tag{45}$$

Let us consider $A(N, n, q, 0, 0)$. Plugging

$$s_{(n)}(q^{-n}, q, \dots, q^{N-1}) = q^{-n^2} + \mathcal{O}(q^{-n^2+1}), \tag{46}$$

into (43) leads to

$$q^{n^2} s_{(n)}(q^{j-1}) s_{(n)}(q^{-n}, q, \dots, q^{N-1}) = q^{n^2} \begin{bmatrix} N+n-1 \\ n \end{bmatrix} \left( q^{-n^2} + \mathcal{O}(q^{-n^2+1}) \right) = 1 + \mathcal{O}(q), \tag{47}$$

where we use the aforementioned fact that $q$-binomials are of the form $1 + \mathcal{O}(q)$. When $r = s \neq 0$, the calculation is slightly more involved. First, we read off from (22) that

$$s_{(n-r,1^r)}(q^{j-1}) = q^{r(r+1)/2}\left(1 + \mathcal{O}(q)\right). \tag{48}$$

We then determine the lowest power of $q$ appearing in

$$s_{(n-r,1^r)}(q^{-(n-r)}, 1, \ldots, q^{r-1}, q^{r+1}, \ldots, q^{N-1}) \tag{49}$$

$$= \sum_{\mu,\nu} s_{(n-r,1^r)/\mu}(q^{-(n-r)}) s_{\mu/\nu}(1, \ldots, q^{r-1}) s_\nu(q^{r+1}, \ldots, q^{N-1}), \tag{50}$$

where we used (33) on the right hand side. Consider the case where $\mu = (1^r)$ and $\nu = \varnothing$. This gives $\lambda/\mu = (n-r-1) \times (1)$, so that $s_{(n-r,1^r)/\mu}(q^{-(n-r)}) = q^{-(n-r)^2}$. Further, we have $s_\mu(q, \ldots, q^{r-1}) = q^{r(r-1)/2}$, and $s_\nu = s_\varnothing = 1$. Therefore,

$$s_{(n-r,1^r)}(q^{-(n-r)}, 1, \ldots, q^{r-1}, q^{r+1}, \ldots, q^{N-1}) = q^{-(n-r)^2} q^{r(r-1)/2}(1 + \mathcal{O}(q)). \tag{51}$$

Plugging this into (43) gives

$$A(N, n, q, r, r) = q^{n^2-2nr} q^{r(r+1)/2} q^{-(n-r)^2} q^{r(r-1)/2}(1 + \mathcal{O}(q)) = 1 + \mathcal{O}(q). \tag{52}$$

One can readily check that any other choice of $\mu$ and $\nu$ leads to higher powers of $q$. For example, choosing $\nu = (1)$ increases the power of $q$ by 2, and choosing a different partition for $\mu$ either increases the power of $q$ by $n-r$ or gives zero (when $\ell((n-r,1^r)/\mu) > 1$). This demonstrates equation (43).

Let us now consider the case where $r \neq s$, to derive equation (44). We take $r < s$ without loss of generality. Taking first $r = 0$, we have

$$A(N, n, 0, s, q) = q^{n^2-ns} s_{(n-s,1^s)}(q^{j-1}) s_{(n)}(q^{-(n-s)}, y, z)$$
$$= q^{n^2-ns} q^{s(s+1)/2} q^{-n(n-s)}(1 + \mathcal{O}(q)) = q^{s(s+1)/2}(1 + \mathcal{O}(q)) = \mathcal{O}(q), \tag{53}$$

where $y = (1, \ldots, q^{s-1})$ and $z = (q^{s+1}, \ldots, q^{N-1})$, as before. Lastly, we check the case where $0 \neq s \neq r \neq 0$, choosing again $r < s$ without loss of generality. Following the same procedure that lead to (51), we find

$$s_{(n-r,1^r)}(q^{-(n-s)}, 1, \ldots, q^{s-1}, q^{s+1}, \ldots, q^{N-1}) = q^{-(n-r)^2} q^{r(r-1)/2}(1 + \mathcal{O}(q)), \tag{54}$$

so that this term, too, appears with a positive power of $q$,

$$A(N, n, r, s, q) = q^{((r-s)^2+r+s)/2}(1 + \mathcal{O}(q)). \tag{55}$$

We have thus shown that terms with $r = s$ contribute $1 + \mathcal{O}(q)$, whereas Schur bilinears with $r \neq s$ contribute terms of $\mathcal{O}(q)$. As mentioned above, powers of $q$ which do not contain a factor $N$ cancel out in the sum over $r$ and $s$, leaving only a linear ramp plus terms of the form $q^{(N+\cdots)}$, as can be seen in (40).

The calculations described here have a simple knot-theoretical interpretation. Remember that $A(N, n, r, s, q)$ is proportional to the HOMFLY invariant of a Hopf link, where the two components of the Hopf link carry $U(N)$ respresentations corresponding to partitions $(n-r, 1^r)$ and $(n-s, 1^s)$, respectively. The above results entail that Hopf links of Wilson lines carrying hook-shaped representations only give a contribution of (order) unity for two identical representations. Therefore, in the limit $q \to 0$, where the CSMM reduces to the CUE, all Hopf link invariants with different $n$-box hook-shaped partitions go to zero. On the other hand, those with identical $n$-box hook-shaped partitions go to one. This means, for example, that an unknot carrying $(a, 1^b)$ with $b \neq 0$ has invariant equal to zero, but if we tie two of these unknots together to form a Hopf link the resulting invariant equals one. On the other hand, an unknot carrying representation $(a)$ has invariant equal to one, but if we tie it to an unknot carrying $(a-b, 1^b)$ with $b \neq 0$ to form a Hopf link, the result is again zero.

## 3.2 Relaxing the assumption that $n < N$

If we relax the assumption that $n < N$, the SFF will eventually reach a plateau for large enough $n$. A well-known, heuristic way to see that such a plateau is eventually reached is as follows. If we consider a diagonal matrix $V = \text{diag}(e^{i\phi_1}, e^{i\phi_2}, \ldots, e^{i\phi_N})$ with all $\phi_j$ taking random values in $[0, 2\pi)$, and we average over $\phi_j$, we get

$$\left\langle |\text{tr}V^n|^2 \right\rangle = \left\langle \sum_{k,l=1}^{N} e^{in(\phi_k - \phi_m)} \right\rangle = N. \tag{56}$$

Here, we use the fact that $e^{in(\phi_k - \phi_m)}$ equals 1 for $k = m$, whereas for $k \neq m$ it is a random variable on ($2n$ copies of) the complex unit circle, which averages to zero. A system with eigenphases distributed randomly across the unit circle therefore has a constant SFF, as was mentioned in the introduction. On the other hand, random unitary matrices $U$ display level repulsion with overwhelming probability, so that their eigenvalues tend to distribute more evenly across the complex unit circle. Therefore, $\left\langle |\text{tr}U^n|^2 \right\rangle$ is much lower than $N$ for small values of $n$. However, for $n$ close to $N$, we have that $n$ becomes of the order of the average spacing $\phi_{k+1} - \phi_k$. In that case, $e^{in(\phi_k - \phi_l)}$ is an approximately random element of the complex unit circle for all $k \neq l$, so that these again average to zero and only the constant contribution $N$ coming from $k = l$ remains. This explains the origin of the plateau for $n \gtrsim N$.

Alternatively, the emergence of the plateau can be understood to arise from the fact that $s_\lambda(x) = 0$ for $\ell(\lambda) > |x|$. In particular, the averages appearing on the right hand side of (30), are weighted matrix integrals written in (5) over Schur polynomials of the form $s_{(n-r,1^r)}(U) = s_{(n-r,1^r)}(e^{i\phi_j})$. If $\ell(s_{(n-r,1^r)}) = r + 1 > N$, then $s_{(n-r,1^r)}(U) = 0$. Therefore, only Schur bilinears of the form

$$\left\langle s_{(n-r,1^r)} s_{(n-s,1^s)} \right\rangle, \quad r, s \leqslant N - 1, \tag{57}$$

give a non-zero contribution to (30). It was demonstrated in the previous subsection that, for $r, s \leqslant N - 1$,

$$\left\langle s_{(n-r,1^r)} s_{(n-s,1^s)} \right\rangle = \delta_{r,s} + \mathcal{O}(q), \tag{58}$$

from which arises the linear ramp. From equation (58), it follows that the linear ramp arising from $r = s \leqslant N - 1$, saturates at a plateau for $n = N$. However, one should note that this does not take into account terms of $\mathcal{O}(q^N)$, which will turn out to have a significant impact on the shape of the SFF, delaying the onset of the plateau as one increases $q^N$.

To implement $n > N$ in the expression for the SFF derived at the start of this section, one should take into account that

1. $s_{(e,1^b)}(y) = 0$ for $b > N - d - 2$,

2. $e_{b-g}(y) = 0$ for $g > d + b + 1 - N$,

where $x = q^{-c}$, $y = q^{d+1}, \ldots, q^{N-1}$, and $z = 1, \ldots, q^{d-1}$, as before. Note that the functions above both arise as $s_{\rho/\nu}(y)$ for $\rho = (e, 1^b)$ in equation (33), where $e_{b-g}(y)$ is given in equation (36), whereas $s_{\rho/\nu}(y)$ reduces to $s_{(e,1^b)}(y)$ for $\nu = \varnothing$. We plot $K(n)$ resulting from this calculation for $N = 10$ and $N = 20$ and various choices of and $q$.

As one can see, the SFF is closest to a linear ramp for small values of $q$, which is to be expected as the limit $q \to 0$ corresponds to the CUE. Further, disconnected SFF becomes large for small $n$ as we increase $q$, leading to large, oscillating deviations close to the origin. Comparing figures 1 and 2 reveals that, for fixed $q$, deviation from a linear ramp decreases as we

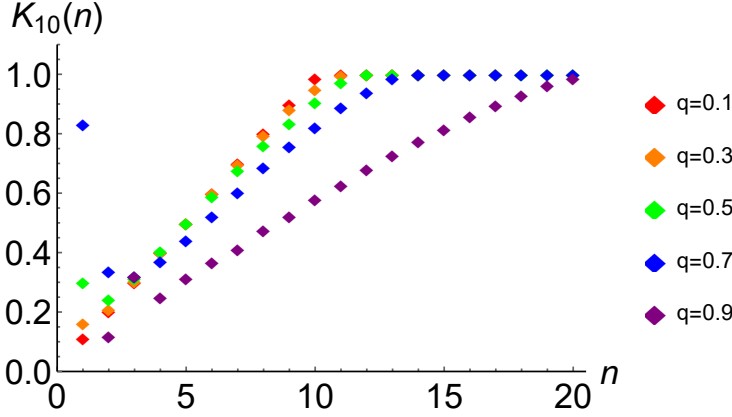

Figure 1: The SFF for $N = 10$ and various values of $q$.

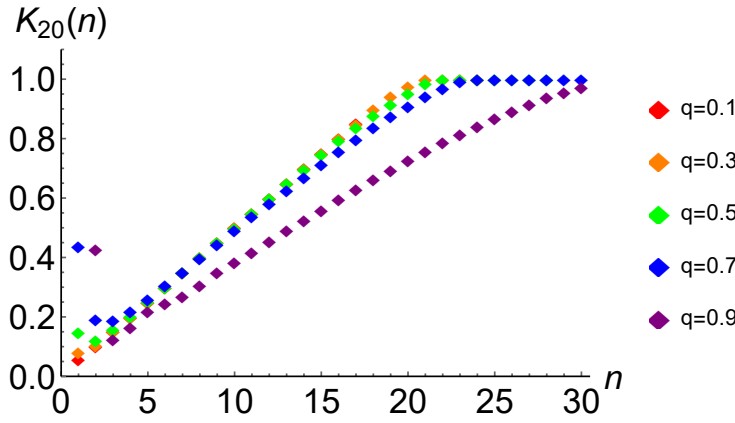

Figure 2: The SFF for $N = 20$ and various values of $q$.

increase $N$. This suggests that a combination of $q$ and $N$ controls the approximate slope away from the origin.

The aforementioned observations that, as we increase $q$, a dip emerges and the slope of the SFF decreases, are not unrelated. In particular, we find that $\sum_{n=1}^{k} F(n)$ for large enough $k \geqslant N$ is almost independent of $q$. That is, as we increase $q$, we get positive contributions to $A(k)$ arising from the disconnected SFF which are compensated by a decrease in the slope of $F(n)$. We define, for $k > N$, the logarithm of the difference between the sum over the SFF of the CSMM and the CUE ($q \to 0$) SFF,

$$A(k) := \log \left[ \sum_{n=1}^{k} F(n) - N^2/2 - N(N - k) \right]. \tag{59}$$

We plot the results for $N = 10$ and $k = 10, \ldots, 20$ below. It is clear that the difference decreases quite rapidly with $k$ until it stabilizes around some small value. Further, we see that the difference decreases more slowly and acquires a larger minimum value as we increase $q$.

The SFF's plotted above were found without unfolding. To see the effect of unfolding, the connected SFF was computed numerically for $N = 10$ and $N = 20$ numerically, using the Metropolis algorithm to generate the spectra. The unfolding is done via Gaussian kernel density estimation using the Silverman rule for bandwidth selection. The data sets for $N = 10$ and $N = 20$ contain at least 10.000 and 5.000 samples, respectively, such that at least 100.000 levels are involved in the calculation of both SFF's. These are plotted below, where, to distinguish them from the analytically calculated (and non-unfolded) SFF's, we denote the numerically

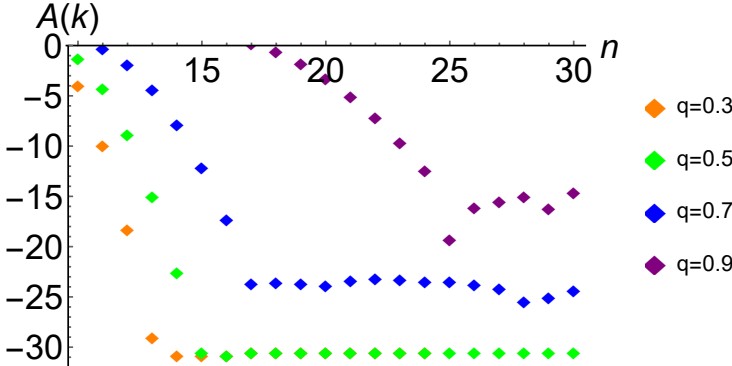

Figure 3: The logarithm of the difference between the sum over the CSMM SFF for $N = 10$ and the CUE ($q \to 0$) SFF, plotted for various values of $q$. We see that the difference is very small and decreases quite rapidly with $k$ but, conversely, increases with $q$.

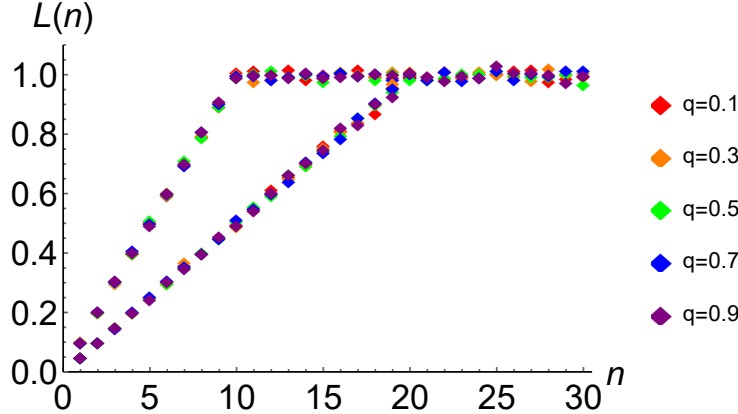

Figure 4: The numerically computed connected unfolded SFF's for $N = 10$ and $N = 20$, which can be distinguished by the fact that they arrive at a plateau at $n = 10$ and $n = 20$, respectively. Note that we plot only the connected SFF here, as opposed to figures 1 and 2, to simplify comparison with Wigner-Dyson universality. It is clear that Wigner-Dyson universality is recovered to high precision for both $N = 10$ and $N = 20$ and all $q$ considered here.

computed SFF's by $L(n)$.[2] We emphasize once more that these are the connected SFF's, where we omit the disconnected part to enable easier comparison with Wigner-Dyson universality. The SFF's for $N = 10$ and $N = 20$ are distinguished by the fact that they arrive at a plateau at $n = 10$ and $n = 20$, respectively. As we can see, for both $N = 10$ and $N = 20$ and all values of $q$ we consider, the connected SFF's exhibit Wigner-Dyson universality rather than intermediate statistics. We will arrive at similar conclusions for the 't Hooft limit below, albeit via different methods.

### 3.3 The SFF in the 't Hooft-limit

Taking the 't Hooft limit, $N \to \infty$ and $g_s \to 0$ such that $t = N g_s =$ finite, leads to $q \to 1$ and $q^N = y$ with $0 \leqslant y < 1$. In this limit, the SFF turns into a remarkable sequence of polynomials of degree $2n - 1$ in $y$. We will first consider these polynomials and their properties, before

---

[2]The data for these plots was kindly provided by Wouter Buijsman.

turning to the question of unfolding. We have calculated the connected SFF for $n = 1, \ldots, 11$, resulting in the expressions below.

$$F(1)_c = 1 - y,$$

$$F(2)_c = 2 - 4y + 6y^2 - 4y^3,$$

$$F(3)_c = 3 - 9y + 36y^2 - 84y^3 + 90y^4 - 36y^5,$$

$$F(4)_c = 4 - 16y + 120y^2 - 560y^3 + 1420y^4 - 1968y^5 + 1400y^6 - 400y^7,$$

$$F(5)_c = 5 - 25y + 300y^2 - 2300y^3 + 10150y^4 - 26880y^5 + 43400y^6$$
$$- 41800y^7 + 22050y^8 - 4900y^9,$$

$$F(6)_c = 6 - 36y + 630y^2 - 7140y^3 + 47880y^4 - 200592y^5 + 544824y^6 - 974160y^7$$
$$+ 1137780y^8 - 834960y^9 + 349272y^{10} - 63504y^{11},$$

$$F(7)_c = 7 - 49y + 1176y^2 - 18424y^3 + 173460y^4 - 1042524y^5 + 4187736y^6$$
$$- 11565624y^7 + 22246686y^8 - 29742020y^9 + 27087984y^{10} - 16024176y^{11}$$
$$+ 5549544y^{12} - 853776y^{13},$$

$$F(8)_c = 8 - 64y + 2016y^2 - 41664y^3 + 522480y^4 - 4237632y^5 + 23380896y^6$$
$$- 90830784y^7 + 253846296y^8 - 515838400y^9 + 762521760y^{10}$$
$$- 810927936y^{11} + 604107504y^{12} - 299065536y^{13} + 88339680y^{14}$$
$$- 11778624y^{15},$$

$$F(9)_c = 9 - 81y + 3240y^2 - 85320y^3 + 1372140y^4 - 14394996y^5 + 103900104y^6$$
$$- 535847400y^7 + 2026445850y^8 - 5713765200y^9 + 12118597920y^{10}$$
$$- 19364383584y^{11} + 23165382240y^{12} - 20414698920y^{13} + 12853423440y^{14}$$
$$- 5468226192y^{15} + 1407913650y^{16} - 165636900y^{17},$$

$$F(10)_c = 10 - 100y + 4950y^2 - 161700y^3 + 3240600y^4 - 42617520y^5 + 388588200y^6$$
$$- 2556668400y^7 + 12488661900y^8 - 46202499200y^9 + 131172321280y^{10}$$
$$- 287919216000y^{11} + 489596250000y^{12} - 642659556000y^{13}$$
$$+ 644511582000y^{14} - 484405727520y^{15} + 263957736900y^{16}$$
$$- 98425126800y^{17} + 22457091800y^{18} - 2363904400y^{19},$$

$$F(11)_c = 11 - 121y + 7260y^2 - 287980y^3 + 7031310y^4 - 113142744y^5$$
$$+ 1269259992y^6 - 10345746840y^7 + 63147440070y^8 - 295025713840y^9$$
$$+ 1071727584928y^{10} - 3059501029728y^{11} + 6907003486240y^{12}$$
$$- 12358366232520y^{13} + 17490417413040y^{14} - 19447530019632y^{15}$$
$$+ 16771920490182y^{16} - 10982054062980y^{17} + 5272925154640y^{18}$$
$$- 1749762036880y^{19} + 358415185128y^{20} - 34134779536y^{21}. \tag{60}$$

To the best of the authors' knowledge, the above polynomials have not appeared in the literature before. Their complicated form belies the fact that the SFF appears to be very close to a straight line for any $y$, with decreasing slope for increasing $y$, see figure 5. We emphasize again that the SFF's plotted in figure 5 were found without applying any unfolding. We will consider the issue of unfolding by rescaling the spectrum in section 3.3.2, see in particular figure 7. In fact, there are three choices of $y$ for which the SFF is a perfectly straight line. Writing $F(n; y)_c$ to indicate dependence on $y$, we have

$$F(n; 0)_c = n,$$

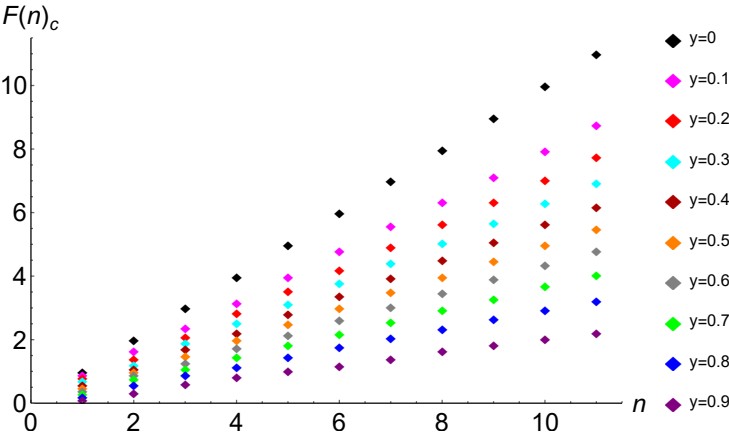

Figure 5: The connected SFF in the 't Hooft limit, without unfolding, written in equation (60). As one can see, the resulting SFF is very close to a straight line for any choice of $y$.

$$F(n;1/2)_c = \frac{n}{2}\,,$$
$$F(n;1)_c = 0\,. \tag{61}$$

The fact that $F(n;0)_c = n$ was already mentioned in section 3.1 and is, in essence, one of the results derived in [3]. The last equality, written as a limit for $y$, can easily be seen to be generally true and will be further commented on in section 3.4. The middle equality, on the other hand, is a priori completely unexpected (at least to the authors).

One can see from the above plot that the SFF appears to be symmetric around $F(n;1/2)_c = \frac{n}{2}$. Indeed, we find that

$$F(n;y)_c + F(n;1-y)_c = n\,. \tag{62}$$

It can be seen that the polynomials $F(n)_c$ appearing in (60) can factorized into the product of a factor $n(1-y)$ and polynomials $p_n(y)$ of degree $2n-2$. The first few of them are given by

$$
\begin{aligned}
p_1 =&\, 1\,,\\
p_2 =&\, 1 - y + 2y^2\,,\\
p_3 =&\, 1 - 2y + 10y^2 - 18y^3 + 12y^4\,,\\
p_4 =&\, 1 - 3y + 27y^2 - 113y^3 + 242y^4 - 250y^5 + 100y^6\,,\\
p_5 =&\, 1 - 4y + 56y^2 - 404y^3 + 1626y^4 - 3750y^5 + 4930y^6 - 3430y^7 + 980y^8\,,\\
p_6 =&\, 1 - 5y + 100y^2 - 1090y^3 + 6890y^4 - 26542y^5 + 64262y^6 - 98098y^7\\
&\, + 91532y^8 - 47628y^9 + 10584y^{10}\,,\\
p_7 =&\, 1 - 6y + 162y^2 - 2470y^3 + 22310y^4 - 126622y^5 + 471626y^6 - 1180606y^7\\
&\, + 1997492y^8 - 2251368y^9 + 1618344y^{10} - 670824y^{11} + 121968y^{12}\,,\\
p_8 =&\, 1 - 7y + 245y^2 - 4963y^3 + 60347y^4 - 469357y^5 + 2453255y^6 - 8900593y^7\\
&\, + 22830194y^8 - 41649606y^9 + 53665614y^{10} - 47700378y^{11} + 27813060y^{12}\\
&\, - 9570132y^{13} + 1472328y^{14}\,.
\end{aligned}
\tag{63}
$$

We were able to identify the coefficients for the following powers of $y$ as

$$
\begin{array}{rcl}
y^0 & : & 1, \\
y^1 & : & -n, \\
y^2 & : & n^2(n+3)/2, \\
y^3 & : & -n(n-1)(2+10n+6n^2+n^3)/6, \\
y^4 & : & n(n-1)(-72-224n-28n^2+87n^3+40n^4+5n^5)/144, \\
y^{2n-1} & : & -(C_n^{2n})^2(2n-1)/2(n+1), \\
y^{2n} & : & (C_n^{2n})^2/(n+1),
\end{array}
$$

where we have the binomial coefficient $C_n^{2n} = (2n)!/(n!)^2$. It seems that no further information about the expansion coefficients of $p_n(y)$ can be obtained easily. This prevents us from generalizing the connected SFF beyond the 11 terms written in (60).

We list here some further observations on these polynomials. We notice that all $p_n(y) - 1$ are divisible by $y(2y-1)$. Denoting $s_n(y) = (p_n(y)-1)/(y(2y-1))$, we find that these can be expressed as functions of $x$, which reduces the degree even further. Finally, considering the polynomials $w_n(y) = s_{n+1}(y) - s_n(y)$ one can realize[3] that they have the following remarkable properties:

a) all the roots of $w_n(y)$ are real;

b) they occupy the interval $[0, 1]$;

c) the roots have the interlacing property, meaning that the roots of lower degree polynomials are located in between the roots of higher degree polynomials.[4]

This and other observations suggest that the polynomials $w_n(y)$ could form a family of orthogonal polynomials. However, the polynomials in (63) are only of even degree, while the odd-degree polynomials are missing. It would be very interesting to extend the sequence of polynomials to terms of higher degree and further explore some of the considerations described above.

### 3.3.1 Level density

We now consider the question of unfolding in the 't Hooft limit, where we have

$$
N^{-1} \langle \mathrm{tr} U^n \rangle = \frac{1}{2n \log y} \left[ P_n(2y-1) - P_{n-1}(2y-1) \right], \tag{64}
$$

where $P_n$ is the Legendre polynomial. This was already found in [54]. In particular, $\langle \mathrm{tr} U^n \rangle$ diverges in the 't Hooft limit in such a way that $N^{-1} \langle \mathrm{tr} U^n \rangle$ is generally finite, as can be seen from taking $q \to 1$ in (38). We thus have a level density which is no longer flat but contains an oscillatory contribution as well,

$$
\rho(\theta; y) = (2\pi)^{-1} \left[ 1 + \frac{1}{\log y} \sum_{n=1}^{\infty} \frac{P_n(2y-1) - P_{n-1}(2y-1)}{n} \cos(n\theta) \right]. \tag{65}
$$

For $|t| < 1$, the generating function of the Legendre polynomials reads

$$
P(x, t) = \sum_{n=0}^{\infty} P_n(x) t^n = \frac{1}{\sqrt{1 - 2xt + t^2}}, \tag{66}
$$

---

[3]We are very grateful to Dr. Denis Kurlov for pointing this fact out to us.

[4]Such interlacing has been observed in related contexts, see [58].

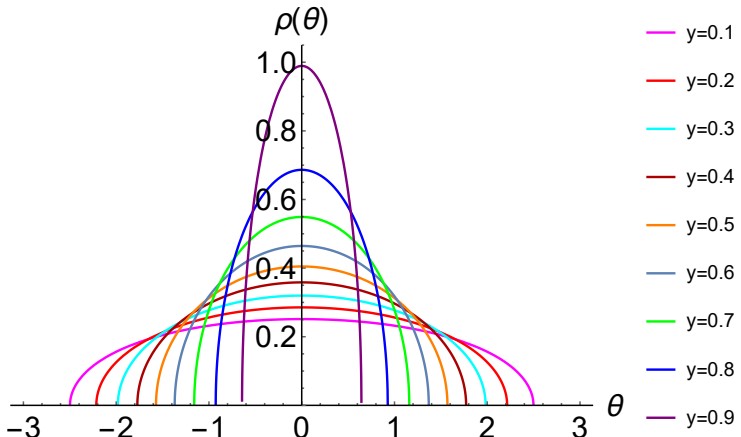

Figure 6: The level density in the 't Hooft limit plotted for various values of $y$. For $y > 0$, a gap opens at $\theta_c = \arccos(2y - 1)$ which increases in size with $y$, with the level remaining a convex function.

which can be integrated to find, for $|z| < 1$,

$$\int_0^z P(x, t) = \sum_{n=1}^{\infty} \frac{P_{n-1}(x)z^n}{n} = \log\left(z - x + \sqrt{1 - 2xz + z^2}\right) - \log(1 - x). \tag{67}$$

Similarly,

$$\sum_{n=1}^{\infty} \frac{P_n(x)z^n}{n} = \log 2 - \log\left(1 - xz + \sqrt{1 - 2xz + z^2}\right). \tag{68}$$

Using Abel's theorem, one may find that

$$\rho(\theta; y) = \frac{1}{2\pi} + \frac{1}{2\pi \log y}\left[\log 2 - \log R(\cos\theta, 2y - 1)\right], \tag{69}$$

where

$$R(z, x) = \begin{cases} (\sqrt{1 + z} + \sqrt{z - x})^2, & x < z, \\ 1 + x, & x \geq z. \end{cases} \tag{70}$$

It is easy to see that the second case listed above, $x \geq z$, gives a zero level density, since then $\log R(z, x) = \log 2y$. In particular, a gap opens in the spectrum,[5]

$$\rho(\theta; y) = 0, \quad \theta > \theta_c = \arccos(2y - 1). \tag{71}$$

This is plotted in figure 6 for $y = .1, .2, \ldots, .9$. One can see that these level densities are approximately of semicircular form. Indeed, writing $r(\theta; y) = \rho(0; y)\sqrt{1 - \frac{\theta^2}{\theta_c^2}}$, the difference between $\rho(\theta; y)$ and $r(\theta; y)$ for $y = 0.1, 0.2, \ldots, 0.9$ remains smaller than $0.006$ for all $\theta$ and decreases with $y$.

---

[5]This critical angle was already found in [54], although our level density appears to be different from the expression obtained there.

### 3.3.2 Unfolding

To compare the connected SFF in the 't Hooft limit with the CUE result, we have to unfold the spectrum. Strictly speaking, unfolding involves a change of variables to the staircase function, (see e.g. section 5.19 of [59]),

$$\sigma(\theta) = \int_{\theta_c}^{\theta} d\theta' \rho(\theta') . \tag{72}$$

The level density in terms of $\sigma$ is a perfectly flat function. The unfolded SFF is then given by

$$\frac{1}{N} \langle | \sum_{j=1}^{N} e^{2\pi i \sigma_j} |^2 \rangle . \tag{73}$$

However, this unfolding procedure is often difficult in practice, and our case is no exception. Finding $\sigma(\theta)$ using the closed form expression for the level density obtained above is not very complicated, but the expression in (73) is not amenable to evaluation. For this reason, we instead perform a constant rescaling to a variable $s$ in terms of which the level density $\rho(s)$ averaged over its support is independent of $q^N$, that is, we simply rescale so that the average spacing is the same for all $y$. For any value of $y$, we take the support of the level density and imagine we can replace the level density by a box-shaped density of the same support. We then rescale the support so that it is again of size $2\pi$. To do so, we write

$$s(\theta) = \frac{\pi\theta}{\theta_c}, \quad s \in [0, 2\pi), \tag{74}$$

so that averaging over its (rescaled) support gives

$$\overline{\rho} = \frac{1}{2\pi} \int_0^{2\pi} ds \rho(s) = \frac{1}{2\pi} = \rho(\theta)_{\text{CUE}} . \tag{75}$$

In terms of the rescaled eigenphases, the level density $\rho(s)$ is of almost exactly the same shape for any $y > 0$, that is, the various densities in 6 are approximately related by rescaling. With this unfolding, the SFF is given by

$$\tilde{F}(n) = \left\langle \left| \sum_{j=1}^{N} e^{2\pi i s_j} \right|^2 \right\rangle = F\left(\frac{\pi n}{\theta_c}\right) . \tag{76}$$

The discrete SFF, $F\left(\frac{\pi n}{\theta_c}\right)$, can only be evaluated at integer $\frac{\pi n}{\theta_c}$. However, we saw in figure 5 that $F(n)_c$ is very close to a linear ramp with slope $\leqslant 1$. Since

$$F(n)_c \approx f(y)n, \quad 0 \leqslant f(y) \leqslant 1, \tag{77}$$

the unfolded connected SFF is approximately

$$\tilde{F}(n)_c \approx \frac{\pi}{\theta_c} F(n)_c =: G(n), \tag{78}$$

which is plotted in 7.

It is clear that $G(n)$ closely resembles a linear ramp of unit slope for all $y$ except close to unity, with resemblance increasing with $n$. This entails that $f(y) \approx \frac{\theta_c}{\pi}$. Only for $y$ close to 1 do we get a significant deviation from WD-universality, with $G(n) \to 0$ for $y \to 1^-$. This demonstrates that the unfolded connected SFF in the 't Hooft limit reproduces WD universality to high precision for $n = 1, \ldots, 11$ and for all $y$ except $y \approx 1$. Although it might be that

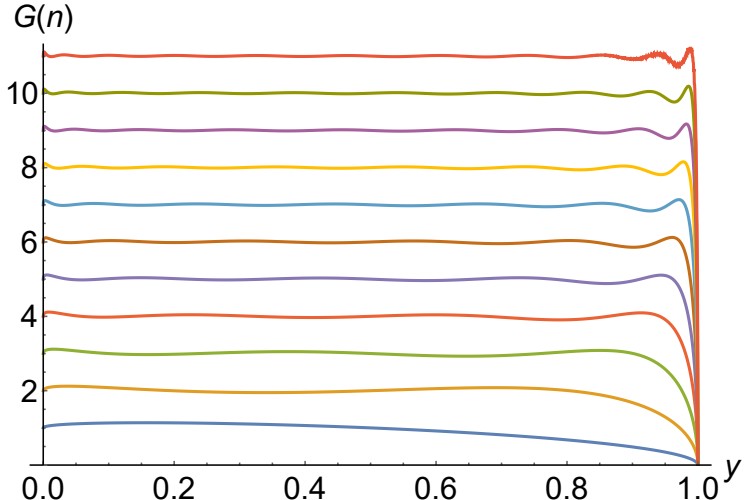

Figure 7: The unfolded connected SFF, $G(n) =: \frac{\pi}{\theta_c} F(n)_c$, for $n = 1, \ldots, 11$. At $y = 0$, we have exactly $G(n) = n$. For $y > 0$, it is clear that $G(n) \approx n$ remains true to high precision, especially for larger $n$, so that the unfolded connected SFF is very close to a linear ramp with unit slope. Only for $y \to 1^-$ does $G(n)$ go to zero and is any resemblance to WD-universality lost.

deviations from WD universality will emerge for lower values of $y$ as we increase $n$ beyond 11, the aforementioned facts that $G(n)$ is close to a linear ramp and that this precision in fact increases with $n$ would appear to render such deviations rather unlikely. Conversely, it may be that deviations from WD will continue to be squeezed into an ever smaller interval below $y = 1$ as we increase $n$, but here, too, we cannot make definitive statements. We note again that the unfolding implemented here involved only a constant rescaling of the eigenphases. Perhaps unfolding as in (73) would remove what deviations from the linear ramp remain in the plot below, but this is of course speculative. Further, spectral form factors of various systems often display non-universal behavior close to the origin, which then disappear further away from the origin where the SFF approaches a linear ramp. The deviations seen in the plot below, except for the region $y \approx 1$, may be just such an effect.

As mentioned previously, the CSMM reduces to the CUE for $q \to 0$, where we know WD-universality to hold. In our previous work [3], we demonstrated that WD-universality holds for any $q < 1$ and $N \to \infty$ as well. The 't Hooft limit, which involves $q \to 1^-$, should then constitute the greatest possible deviation from the CUE result, yet WD-universality reappears for all $y$ other than $y \approx 1$ after even a very simple unfolding. This would appear to be rather unexpected, as the CSMM was introduced and extensively studied as a random matrix model for intermediate statistics, as described in more detail in the introduction. We further comment on this result and its implications in the conclusion.

## 3.4 Non-commutativity of the limit $q \to 1$ and $N \to \infty$

One can see from the expression of the SFF that the limits $q \to 1$ and $N \to \infty$ do not commute. Such non-commutativity has been discussed in the literature already decades ago, see [54]. In particular, if we take $q \to 1$ into expression (16) for finite $N$, the Schur polynomials simply give the dimension of the representation, that is

$$s_\lambda(1, 1, \ldots, 1) = \dim \lambda. \tag{79}$$

Plugging this into (16) with $\lambda = (n - r, 1^r)$ and $\mu = (n - s, 1^s)$ shows that the SFF would be simply given by

$$\left( \sum_{r=0}^{n-1} (-1) \frac{(N + n - r - 1)!}{(N - r - 1)!(n - r - 1)!r!n} \right)^2 = N^2, \tag{80}$$

for all $n$. In terms of knot theory, we see that taking $q \to 1$ for $N$ finite breaks the $(2n, 2)$-torus link that is the SFF up into its separate $(n, 1)$-torus knot components, as we have

$$\lim_{q \to 1} \langle |\mathrm{tr} U^n|^2 \rangle = \lim_{q \to 1} \langle \mathrm{tr} U^n \rangle \langle \mathrm{tr} U^{-n} \rangle = \lim_{q \to 1} \left( \langle \mathrm{tr} U^n \rangle \right)^2. \tag{81}$$

The connected SFF then equals zero, as was the case in section 3.3. We consider the case where $t = N g_s \ll 1$ is very small. This allows us to use the following expansion [ [57] I.3, example 10],

$$s_\lambda(1 + x_1, 1 + x_2, \ldots, 1 + x_N) = \sum_\mu d_{\lambda\mu} s_\mu(x_1, \ldots, x_N), \tag{82}$$

where we sum over all $\mu \subseteq \lambda$ and where

$$d_{\lambda\mu} = \det \begin{pmatrix} \lambda_i + n - 1 \\ \mu_j + n - j \end{pmatrix}_{1 \leqslant i, j \leqslant N}. \tag{83}$$

Some simple examples are given by (see e.g. [60])

$$d_{\lambda\varnothing} = \dim \lambda, \quad d_{\lambda\square} = \dim \lambda \frac{c_1(\lambda)}{N}, \tag{84}$$

where the first Casimir invariant is given by $c_1(\lambda) = |\lambda| = \sum_i \lambda_i$. For $q = e^{-g_s}$ close to 1 and $N g_s \ll 1$, we have

$$s_{(a, 1^b)}(q^{j-1}) \simeq s_{(a, 1^b)}(1, 1 - g_s, 1 - 2g_s, \ldots) = \sum_\mu d_{(a, 1^b)\mu} s_\mu(0, -g_s, -2g_s, \ldots). \tag{85}$$

Expanding up to linear order in $g_s$, we only get contributions for $\mu = \varnothing$ and $\mu = \square$, which gives

$$
\begin{aligned}
s_{(a, 1^b)}(0, -g_s, -2g_s, \ldots) &= \dim(a, 1^b) \left( 1 + \frac{a + b}{N}(-g_s - 2g_s - \ldots) \right) \\
&= \dim(a, 1^b) \left( 1 - \frac{(a + b)(N - 1)}{2} g_s \right). \tag{86}
\end{aligned}
$$

Further,

$$
\begin{aligned}
s_{(a, 1^b)}(q^{-c}, 1, q, \ldots, q^{d-1}, q^{d+1}, \ldots, q^{N-1})) &\simeq s_{(a, 1^b)}(c g_s, -g_s, \ldots, -(d - 1)g_s, -(d + 1)g_s, \ldots) \\
&= \dim(a, 1^b) \left[ 1 + (a + b) \left( \frac{1 - N}{2} + \frac{c + d}{N} \right) g_s \right]. \tag{87}
\end{aligned}
$$

Plugging this into (16) for $\lambda = (n - r, 1^r)$ and $\mu = (n - s, 1^s)$ gives

$$\langle W_{\lambda\mu} \rangle - \langle W_\lambda \rangle \langle W_\mu \rangle = \dim(n - r, 1^r) \dim(n - s, 1^s) \frac{n^2 g_s}{N} + \mathcal{O}(g_s^2). \tag{88}$$

We thus see that, to first order in $N g_s$, the Wilson loop factorizes, so that

$$F(n)_c = t n^2 + \mathcal{O}(t^2). \tag{89}$$

If we now take $N \to \infty$ in such a way that $t$ remains small, we clearly get a very different result from the linear ramp $F(n)_c = n$ that is found when taking $q \to 1$ after $N \to \infty$. Indeed, one may check that the connected SFF in the 't Hooft limit for small $n$ and $y \lesssim 1$ is very close to $t n^2$.

# 4 Conclusion

In this work, we calculate the SFF of the CSMM for general $q = e^{-g_s}$ and matrix size $N$. We find that, as $y = q^N \to 0$, we recover our results in [3] for the case of the CSMM. That is, the connected SFF, $F(n)_c = \langle |\mathrm{tr} U^n|^2 \rangle - \langle \mathrm{tr} U^n \rangle^2$, is exactly given by a linear ramp of unit slope. For $y$ different from zero, we see that the (approximate) slope of the SFF becomes smaller than one as we increase $q$. For $n > N$, the SFF eventually saturates at a plateau, which takes longer for larger $q$ due to the fact that the slope decreases with $q$. The emergence of the linear ramp and its saturation at a plateau is shown to arise from the properties of Schur bilinears appearing in the character expansion of the SFF.

In the 't Hooft limit, the connected SFF reduces to a sequence of polynomials of degree $2n - 1$ in $y$, which we calculated up to $n = 11$. As far as the authors are aware, this sequence has not appeared in the literature thus far. Before unfolding, the connected SFF is again approximately given by a linear ramp, but now with slope $\leqslant 1$. As described in section 3.3, the SFF is symmetric around $y = 1/2$, where we have $F(n)_c = \frac{n}{2}$. Further, $N^{-1} \langle \mathrm{tr} U^n \rangle$ is generally finite, so that the level density is no longer flat. In particular, a gap opens up for $y > 0$, so that the support of the level density is given by $(-\theta_c, \theta_c)$ for $\theta_c = \arccos(2y - 1)$. To unfold the spectrum, we rescale the eigenphases so that the support is again an interval of length $2\pi$. Upon unfolding, the SFF is again very close to a linear ramp for all $y$ except $y$ close to 1, with precision increasing with $n$. That is, we recover Wigner-Dyson universality even in the 't Hooft limit as long as $y$ is not too close to unity. It would be interesting to see whether deviations from WD for $y \to 1^-$ persist in $F(n)_c$ for $n \geqslant 12$, whether they continue to be squeezed in a smaller interval below $y = 1$, or whether they perhaps disappear altogether. The fact that we have to base our conclusions on 11 instances of the connected SFF prevents us from making definitive statements on this point, but perhaps future investigation will shed further light on it.

As described in the introduction, the CSMM was originally introduced [1] to describe the intermediate statistics of disordered electrons at the mobility edge, and there is a significant amount of literature on this application of the CSMM and related ensembles, see e.g. [31, 32, 36, 37, 61–63]. Indeed, the 't Hooft limit involves $q \to 1$, which is the opposite extreme of the CUE limit, $q \to 0$, so it should come as a surprise that WD-universality is recovered even here, and indeed for all $y$ except $y \approx 1$. We emphasize that we do not mean to cast doubt on the results derived in the aforementioned papers. Although their results may appear hard to reconcile with ours at first sight, their analysis involves a different set of limits and approximations, the approximation in our case being the unfolding by a constant rescaling and in their case e.g. considering only a small eigenvalue window around the origin. Their analysis centers on the hermitian version of the CSMM where the weight function is $\sim e^{-\alpha \log^2(x)}$ for $x \in \mathbb{R}$, and it employs the theory of orthogonal polynomials rather than the character expansion used here. The latter two points are not of fundamental importance as they should be merely different descriptions of the same underlying structure, yet they do complicate the comparison between these sets of results. Further, it was shown very recently [64] that defining the SFF as $K(t)$ with $t \in \mathbb{C}$ and taking $\mathrm{Im}(t) \to 0$ generally leads to different SFF than when $t$ is taken to be always real-valued, so it is clear that we have not exhausted the different limits in which one can calculate the SFF. It would be very interesting to understand under what conditions the CSMM or related models exhibit deviations from WD-universality, as apparently it is not enough to take the 'disorder parameter' $q$ to its maximum value.

The authors believe that some of the results derived here are also of mathematical interest. As mentioned in the introduction, the SFF is proportional to the HOMFLY invariant of a $(2n, 2)$-torus link with components carrying fundamental and antifundamental representations. The calculation of the SFF thus provides explicit expressions for new link invariants, both for gen-

eral $q$, $N$ as well as in the 't Hooft limit. Due to the appearance of $U(N)$ Chern-Simons theory at large $N$ in the form of various topological string theories described in the the introduction, and the relation of these large $N$ dualities to enumerative geometry and intersection theory, the results derived here could be of mathematical interest beyond knot theory.

## Acknowledgements

We are greatly indebted to Wouter Buijsman for helping us out with the numerical calculations in this work. Further, we are grateful to Aleksandr Garkun, Vladimir Kravtsov, and Denis Kurlov for useful discussions and for their help at different stages of this project. This work is part of the DeltaITP consortium, a program of the Netherlands Organization for Scientific Research (NWO) funded by the Dutch Ministry of Education, Culture and Science (OCW).

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
