# Peer review of "The spectral form factor in the `t Hooft limit -- Intermediacy versus universality"

_SciPost Physics, doi:SciPost Phys. Core 5, 051 (2022)_

## Round 1 · Referee Report · Anonymous (Referee 1) · 2022-2-21

Strengths

  1. It is a complicated calculation which is correct as far as I can see.

Weaknesses

  1. The main point of the paper the results give new universality class for spectral is not justified.
  2. The main results of the paper are checked by numerical simulations, which can be easily done for this model.
  3. The paper is written for a narrow group of experts in Chern-Simons RMT and will be inaccessible for most of the quantum chaos community.
  4. The paper is just a calculation without physical understanding of the results.

Report

In this paper the authors consider a random matrix model for U(N) Chern-Simons theory on a three dimensional
sphere and calculate the spectral form factor of this model. The authors claim that the level statistics
of model is intermediate between the universal are RMT result and an integrable limit that is not described.
They even claim that it has a novel RMT universality. Below I will argue that even though the
calculation is correct, this conclusion is unjustified, so that the paper cannot be published in its current form.
To convince the reader and the referee, the authors will have show explicitly that the spectral statistics
are in a novel universality class and are not due to for example unfolding issues.

The first issue with the paper is that the authors never give an explicit expression for the partition
function. This might be clear for a narrow audience, but the authors aim at a wider audience. Most likely
the partition function is given by
$$
Z = \int dU \prod_{k=0}^\infty \det (1+q^{k+1/2} U) \det (1+q^{k+1/2} U^\dagger),
$$
where $dU$ is an integral over the Haar measure of U(N) and $q$ is a parameter of the model. If $q=0$
we have the Circular Unitary Ensemble. Please add this formula to the paper. Also state that the
expectation values , $\langle \cdot \rangle$ are with respect to this partition function.

The authors calculate (in fact already in their previous paper) $\langle {\rm tr} U \rangle$ and find
$$
\langle {\rm tr} U \rangle = \frac{(1-q^N \sqrt q} {1-q}.
$$
This vanishes for $q=0$ which is correct because for $q=0$ the eigenvalues are distributed
uniformly alonog the complex unit circle. For $q\ne 0$ the expectation value is nonzero which means
that the eigenvalues are no longer distributed uniformly over the unit circle. In fact the spectrum is
polarized so that the distribution of the eigenphases is no longer uniform, and in order to determine
the universality class of the spectrum from the form factor, the spectrum has to be unfolded. The current
form factor looks exactly like the deviation from GUE one obtains when the spectrum is not unfolded.
Alternatively, the authors could evaluate spectral statistics that are not sensitive to unfolding such as
spacing ratios or the number variance.

The simplest way to address these question is to simulate the partition function numerically at finite $N$.
Since the measure of the above partition function is positive definite, an ensemble with its integrand
as probability distribution can easily be generated by a Metropolis algorithm. Alternatively, the authors
could calculate the spectral density by averaging over several million spectral of CUE matrices (the
CUE distribution is easily accessible on Mathematical, and for $N=10$ it takes only a few minutes
to generate a million spectral (Actually I did that to check the above expression for the partition function).
Then the determinantal factor is introduced by reweighting.

In order to recommend publication of this paper, the authors have to calculate the form factor of
the $unfolded spectrum$. If eigenvalue density is not uniform the form factor will not have the GUE form.
If the authors find a way to do this analytically instead of numerically this
is fine too, but I think it is much simpler and to perform numerical simulations which gives you also easy access
to other spectral observables.

The author claim that the polynomials in (58) represent a novel type of random matrix universality. Such
claims could only be made if the average spectral density is uniform, but it is not. The ’t Hooft limit
of $\langle {\rm tr } U \rangle$ is given by
$$
\frac 1N \langle {\rm tr} U \rangle = \frac {1-y}{\log 1/y}.
$$
This formula should be added to the paper. For $y \to 0$ the expectation value vanishes so
that the spectral statistics is in the CUE class in agreement with (58) of the paper. Universality can only
arise for statistics of eigenvalue correlations in terms of units of the average level spacing. This is
not the case for the discrete form factor as $n$ is independent of $N$ in the ’t Hooft limit. Please
add a discussion of the meaning of universality in this case. To claim universality, the authors have
to show that the expressions in (58) are valid for a wide class of systems and not just the Chern-Simons
RMT, and they have to eliminate the effect due to unfolding.

More generally, integrable systems have never universal eigenvalue statistics. Each integrable system is special.
What is universal is the Dyson Brownian motion $process$ that describes the
transition between an integrable systems and RMT. However, the spectral statistics of each
process depends on the initial condition. So universality can only be used with the proper
disclaimers for intermediate statistics.

Below are several minor corrections:

p2. The authors put quotations around “universal” for this semi-circle which is correct because
a semicircle has never been observed in a physical system. However, they also use quotations
around universal for the GUE spectral form factor. This is incorrect. GUE spectral correlations
have been observed in many different systems, and this universality is very strong, perhaps
even stronger that the universality of critical exponents at a second order phase transition.

The authors use a comma instead of a point to denote decimals (e.g. 0,3 instead of 0.3). This
is clear for Dutch readers, but is confusing for many others who are used to the physics notation
to use a point for decimals. Please correct all figures.

Please add a reference to the paper where the CUE form factor was first calculated. According to
the best of my knowledge it is the paper by Haake, Kus, Sommers, Schomerus and
Zyczkowski,, J. Phys. A: Math. Gen. 29 3641 (1996), but Madan Lal Mehta certain knew the result earlier
than that, but I do not know if he published it.

Also the references on intermediate statistics are quite incomplete. Please check ref. [26] to make sure that you cited the original literature.

On p. 3 the authors mentions ergodic to non-ergodic transitions. However, ergodic systems do not
necessarily have RMT statistics. They also have to be chaotic. Please rephrase.

In conclusion, I cannot publication of this paper in its present form, but I am happy to consider after
the paper has been improved according to the above. In particular, it is essential that the authors
perform numerical simulations to establish the main conclusions of the paper.

Requested changes

There are many changes which are discussed in detail in the
report.

  • validity: low
  • significance: low
  • originality: ok
  • clarity: low
  • formatting: acceptable
  • grammar: excellent

Author:  Ward Vleeshouwers  on 2022-05-17  [id 2485]

(in reply to Report 1 on 2022-02-21)

We respond point-by-point to the comments and objections raised by the referee. We have indicated the statements by the referee with "R" and our answers with "A".

R: In this paper the authors consider a random matrix model for U(N) Chern-Simons theory on a three dimensional sphere and calculate the spectral form factor of this model. The authors claim that the level statistics of model is intermediate between the universal are RMT result and an integrable limit that is not described. They even claim that it has a novel RMT universality. Below I will argue that even though the calculation is correct, this conclusion is unjustified, so that the paper cannot be published in its current form. To convince the reader and the referee, the authors will have show explicitly that the spectral statistics are in a novel universality class and are not due to for example unfolding issues.

A: Please note that we did not claim a novel RMT universality, we merely hinted at its possibility. However, after this question from our referee came in we started to look much closer into the issue of unfolding. Results of our analysis are summarized below. This essentially leads us to the opposite conclusion, namely that this model is in the Wigner-Dyson (WD) universality class, except for some limiting case of $y\approx 1$, please see below.

R: The first issue with the paper is that the authors never give an explicit expression for the partition function. This might be clear for a narrow audience, but the authors aim at a wider audience. Most likely the partition function is given by
\begin{equation}
Z=\int dU \prod_{k=0}^\infty \det (1+q^{k+1/2}U)\det (1+q^{k+1/2}U^\dagger)
\end{equation}
where $dU$ is an integral over the Haar measure of $U(N)$ and $q$ is a parameter of the model. If $q=0$ we have the Circular Unitary Ensemble. Please add this formula to the paper. Also state that the expectation values , $\langle \dots\rangle$ are with respect to this partition function.

A: These expressions should indeed have been added, we do so in the revision.

R: The authors calculate (in fact already in their previous paper) $\langle \text{tr } U \rangle$
and find
\begin{equation}
\langle \text{tr } U \rangle =\frac{(1-q^N)\sqrt{q}}{1-q}~.
\end{equation}
This vanishes for $q=0$ which is correct because for $q=0$ the eigenvalues are distributed uniformly alonog the complex unit circle. For $q\neq 0$ the expectation value is nonzero which means that the eigenvalues are no longer distributed uniformly over the unit circle. In fact the spectrum is polarized so that the distribution of the eigenphases is no longer uniform, and in order to determine the universality class of the spectrum from the form factor, the spectrum has to be unfolded. The current form factor looks exactly like the deviation from GUE one obtains when the spectrum is not unfolded. Alternatively, the authors could evaluate spectral statistics that are not sensitive to unfolding such as spacing ratios or the number variance.

The simplest way to address these question is to simulate the partition function numerically at finite $N$. Since the measure of the above partition function is positive definite, an ensemble with its integrand as probability distribution can easily be generated by a Metropolis algorithm. Alternatively, the authors could calculate the spectral density by averaging over several million spectral of CUE matrices (the CUE distribution is easily accessible on Mathematical, and for $N=10$
it takes only a few minutes to generate a million spectral (Actually I did that to check the above expression for the partition function). Then the determinantal factor is introduced by reweighting.

In order to recommend publication of this paper, the authors have to calculate the form factor of the unfoldedspectrum. If eigenvalue density is not uniform the form factor will not have the GUE form. If the authors find a way to do this analytically instead of numerically this is fine too, but I think it is much simpler and to perform numerical simulations which gives you also easy access to other spectral observables.

The author claim that the polynomials in (58) represent a novel type of random matrix universality. Such claims could only be made if the average spectral density is uniform, but it is not. The ’t Hooft limit of $ \langle \text{tr} U \rangle$ is given by
\begin{equation}
\frac{1}{N} \langle \text{tr} U \rangle = \frac{1 - y}{\log 1/y}.
\end{equation}

This formula should be added to the paper. For $ y \to 0$ the expectation value vanishes so that the spectral statistics is in the CUE class in agreement with (58) of the paper. Universality can only arise for statistics of eigenvalue correlations in terms of units of the average level spacing. This is not the case for the discrete form factor as $n$ is independent of $N$ in the ’t Hooft limit. Please add a discussion of the meaning of universality in this case. To claim universality, the authors have to show that the expressions in (58) are valid for a wide class of systems and not just the Chern-Simons RMT, and they have to eliminate the effect due to unfolding.

A: As stated elsewhere in the paper, we intended to return to the question of universality in a future work, but we agree in retrospect that this point, in particular the issue of unfolding, should have been considered in this paper. This is what we did now:

As pointed out by the referee, we have $\frac{1}{N} \langle \text{tr} U \rangle \neq 0$ in the `t Hooft limit. In fact we have $\frac{1}{N} \langle \text{tr} U^n \rangle \neq 0$ for general $n$. We add the explicit expression for $\frac{1}{N} \langle \text{tr} U^n \rangle$ to the revision. This indeed leads to a level density which is not uniform. In particular, a gap opens in the spectrum at a critical angle $\theta_c = \arccos(2y-1)$. The level density $\rho(\theta)$ remains a convex function for all $y$. We have attached a plot of the densities for various values of $y$. Ideally, one would like to unfold by using
\begin{equation}
\sigma(\theta)= \int_{\theta_c}^{\theta} d\theta' \rho(\theta')~,
\end{equation}
and defining the SFF as
\begin{equation}
\frac{1}{N}\langle |\sum_{j=1}^N e^{2\pi i \sigma_j }|^2 \rangle~.
\end{equation}
However, we do not know how to implement this procedure in the present calculation. For this reason, we instead perform a constant rescaling to a variable $s$ in terms of which the level density $\rho(s)$ averaged over its support is independent of $q^N$. That is, we write
\begin{equation}
s(\theta)=\frac{\pi \theta}{\theta_c} ~ ,~~ s \in [0,2\pi)~,
\end{equation}
so that
\begin{equation}
\overline{\rho} = \frac{1}{2\pi} \int_{0}^{2\pi} ds \rho(s) = \frac{1}{2\pi}= \rho(\theta)_{\text{CUE}} ~.
\end{equation}
So, we essentially replaced smooth curves on the plot by the boxes, such that the area under the box is equal to the area under the smooth density curve. The unfolded SFF (after rescaling by a factor $N$, as in the paper) is then given by
\begin{equation}
\tilde{F}(n) = \langle |\sum_{j=1}^N e^{2\pi i s_j }|^2 =\langle |\sum_{j=1}^N e^{2\pi i s_j }|^2 \rangle =F\left( \frac{\pi n}{\theta_c}\right)~.
\end{equation}
As in the paper, we write the connected SFF as
\begin{equation}
F(n)_c= \langle | \text{tr} U^n |^2 \rangle - (\langle \text{tr} U^n \rangle)^2~.
\end{equation}
The discrete SFF, $F\left( \frac{\pi n}{\theta_c}\right)$, can only be evaluated at $\frac{\pi n}{\theta_c}$ integer. However, as noted in our paper, $F(n)_c$ is very close to a linear ramp with slope $ \leq 1$, with oscillations around it in an envelope that decays with $n$. Since
\begin{equation}
F(n)_c \approx f(y)n~, ~~0\leq f(y) \leq 1~,
\end{equation}
the unfolded connected SFF is approximately
\begin{equation}
\tilde{F}(n)_c \approx \frac{\pi}{\theta_c} F(n)_c~.
\end{equation}
When we plot this for various values of $y$, we recover the linear ramp of unit slope to very high precision, with precision increasing with $n$. Only for $y\to 1^-$ do we have get $ \frac{\pi}{\theta_c} F(n)_c \to 0$. One way to express the precision is to consider
\begin{equation}
\Gamma(n;y) = F(n;y)_c -\frac{\theta_cn}{\pi}~,
\end{equation}
which reaches its maximum value for $n=1,\dots,11$ and $0\leq y \leq 1$ at $\Gamma(1,y^*) =1-y^* -\frac{\arccos(2y^* -1)}{\pi} \lessapprox 0.106, ~y^* = \frac{1 - \sqrt{1 - 4/\pi^2}}{2}~. $
$\Gamma(n;y)$ decreases quite rapidly with $n$, the maximum value of $\Gamma(11;y) $ being smaller than $0.03$.

To our surprise, we find that the connected SFF in the `t Hooft limit is very close to a linear ramp with unit slope for all $y$ except $y\approx 1$, even upon this simple unfolding. It appears we recover Wigner-Dyson universality to very high precision, so it is indeed not correct to state that there might be a novel RMT universality in the `t Hooft limit, at least for all $y$ except $y\approx 1$. We added the analysis above (in more detail) to the revision. Further, we removed statements about the possibility of a novel RMT universality from the paper and add a description of the recovery of Wigner-Dyson universality upon unfolding.

The referee suggested that we perform the unfolding by using numerical methods. However, we find the above analytical calculations convincing in establishing that the connected SFF reproduces the CUE result (linear ramp of unit slope) to high precision. Moreover, performing numerical simulations for finite $N$ will not accurately describe the `t Hooft limit, where $N\to \infty$. This can be found by comparing the SFF for finite $N$ and in the `t Hooft limit for the same values of $q^N$, which was done by the authors but not included in the paper. For these reasons, we would like to focus our revision on the analytical arguments presented above instead of a numerical calculation.

R: More generally, integrable systems have never universal eigenvalue statistics. Each integrable system is special. What is universal is the Dyson Brownian motion process that describes the transition between an integrable systems and RMT. However, the spectral statistics of each process depends on the initial condition. So universality can only be used with the proper disclaimers for intermediate statistics.

A: We are not entirely sure what the referee means by that statement. It is a commonly held notion by now (which is of course a conjecture by Berry and Tabor) that quantum integrable systems have Poissonian level statistics. For a more detailed discussion of this and related points, we would like to refer to the paper by J.-S. Caux and J. Mossel, 1012.3587. We are using the definition QI:ELS from this paper which we believe to be standard in the RMT context.

R: Below are several minor corrections:

p2. The authors put quotations around “universal” for this semi-circle which is correct because a semicircle has never been observed in a physical system. However, they also use quotations around universal for the GUE spectral form factor. This is incorrect. GUE spectral correlations have been observed in many different systems, and this universality is very strong, perhaps even stronger that the universality of critical exponents at a second order phase transition.

The authors use a comma instead of a point to denote decimals (e.g. 0,3 instead of 0.3). This is clear for Dutch readers, but is confusing for many others who are used to the physics notation to use a point for decimals. Please correct all figures.

Please add a reference to the paper where the CUE form factor was first calculated. According to the best of my knowledge it is the paper by Haake, Kus, Sommers, Schomerus and Zyczkowski,, J. Phys. A: Math. Gen. 29 3641 (1996), but Madan Lal Mehta certain knew the result earlier than that, but I do not know if he published it.

Also the references on intermediate statistics are quite incomplete. Please check ref. [26] to make sure that you cited the original literature.

On p. 3 the authors mentions ergodic to non-ergodic transitions. However, ergodic systems do not necessarily have RMT statistics. They also have to be chaotic. Please rephrase.

A: We implemented the changes suggested above.

R: In conclusion, I cannot publication of this paper in its present form, but I am happy to consider after
the paper has been improved according to the above. In particular, it is essential that the authors
perform numerical simulations to establish the main conclusions of the paper.

Attachment:

rdpl.pdf

---

## Round 1 · Referee Report · Anonymous (Referee 2) · 2022-3-31

Report

This is an interesting work which focusses on the evaluation and exhaustive study of the spectral form factor of random matrix ensembles of the intermediate type (characterized by weight functions that have associated q-deformed orthogonal polynomials).

The analytical evaluation seems sound and it is exhaustive. It is not a simple calculation. For example, the trick of using Equation 31 is quite novel and crucial. They also combine non trivial previous analytical results, which are not that well known. However, it can be argued that part of this was already
accomplished in their previous work, already published here.

There are also quite a few side developments that while interesting (and some of them, I found quite interesting and were previously unknown, like the results and discussion in 3.3.1.) may fall a bit short (no doubt due to the difficulty) and do not fully move the discussion in a coherent fashion. Case in point, the above mentioned 3.3.1.: it is quite interesting and novel but has no continuity. It seems difficult that their sort of empirical or top-down method employed there could perhaps be pushed much further. There are quite a few mathematical results on the asymptotics of the Stieltjes-Wigert or Rogers-Szego polynomials that perhaps could help in their very interesting but non-trivial goal of studying correlations in the 't Hooft limit.

I do not agree however, that theirs is just a result on "Chern-Simons matrix models". The model is of broader interest, even if that is perhaps not fully appreciated. One interesting aspect of the whole random matrix formulation of Chern-Simons theory, which has not been described elsewhere, is that somehow gives a sort of common ground between several approaches in quantum topology: It is well known that quantum topology can be equivalently approached in seemingly very different manners (and certainly, in many mathematical circles Chern-Simons theory is not precisely the preferred one). Notice the similarity or even equivalence of the methods and objects studied in this paper and the previous one (and some references therein) and the results by the knot theory group of H.R. Morton, which uses skein theory. See for example https://arxiv.org/pdf/math/0108011.pdf (especially Section 4). While the starting point is very different both approaches end up requiring evaluations of minors of structured matrices with Gaussian entries. This work handles this type of evaluations quite well, with new mathematical results.

The same minors appear in very different, and physically relevant, contexts, see for example the expression (93) in https://arxiv.org/pdf/2103.02545.pdf and the discussion around it, for another take on the interest in computing these minors in specific starting and ending configurations (the representations), precisely the kind of computation carried out here.

The Author's evaluation is also the evaluation of a Homfly polynomial of a torus link (as clearly explained by the authors) and, at least for torus knots, its homfly polynomial can be written in powers of a variable x=sqrt(q)-1/sqrt(q) and the corresponding coefficients have independent interest. In particular, for large N, the leading coefficient is $p_{0}$ and it is a polynomial invariant itself, relevant in the study of periodic knots (studied by P. Traczyk and others). Taking also q->1 this object is a Wilson loop invariant in a 2d YM theory. This is speculative, but I wonder if, in their double scaling limit, and for the torus link invariant studied here, any of these considerations would apply ? If so, maybe that would be a further application of their analytical results.

The physical consequences of their detailed SFF analysis are perhaps unclear (to this reader at least) and I would say that, in spite of the obvious efforts by the authors, the paper is overall a bit difficult to read, no doubt due to its interdisciplinarity while also being quite technical in parts of it.

Since one of the difficulties with this paper is that, content-wise, it may be quite hard for the intended audience, together with the difficulties in advancing the somewhat elusive physics of these intermediate ensembles (beyond their exact appearances, above mentioned, and in Chern-Simons theory), that suggests that perhaps it would be a better fit in a mathematical physics journal, where it could be a very good contribution.
  • validity: good
  • significance: ok
  • originality: ok
  • clarity: low
  • formatting: -
  • grammar: -

Author:  Ward Vleeshouwers  on 2022-05-17  [id 2484]

(in reply to Report 2 on 2022-03-31)

We respond point-by-point to the comments and objections raised by the referee. We have indicated the statements by the referee with "R" and our answers with "A".

R: This is an interesting work which focusses on the evaluation and exhaustive study of the spectral form factor of random matrix ensembles of the intermediate type (characterized by weight functions that have associated q-deformed orthogonal polynomials).

The analytical evaluation seems sound and it is exhaustive. It is not a simple calculation. For example, the trick of using Equation 31 is quite novel and crucial. They also combine non trivial previous analytical results, which are not that well known. However, it can be argued that part of this was already
accomplished in their previous work, already published here.

There are also quite a few side developments that while interesting (and some of them, I found quite interesting and were previously unknown, like the results and discussion in 3.3.1.) may fall a bit short (no doubt due to the difficulty) and do not fully move the discussion in a coherent fashion. Case in point, the above mentioned 3.3.1.: it is quite interesting and novel but has no continuity. It seems difficult that their sort of empirical or top-down method employed there could perhaps be pushed much further. There are quite a few mathematical results on the asymptotics of the Stieltjes-Wigert or Rogers-Szego polynomials that perhaps could help in their very interesting but non-trivial goal of studying correlations in the 't Hooft limit.

A: We agree with the assessment by the referee. We combined what was previously section 3.3.1 with the other mathematical considerations of the SFF in the `t Hooft limit. We realize that there may have been a lack of continuity, but this is largely due to the fact that we only computed a finite number of these polynomials. We emphasize the latter point further in the revision of the paper to make this treatment more self-contained.

With regards to the comments on Stieltjes-Wigert and Rogers-Szegö polynomials, indeed some similar limits have been treated e.g. in 1312.5848v2. However, to apply these to the calculation of the SFF using these results is a non-trivial problem, not in the least because of the fact that one has to compute a complicated double Fourier transform. We agree that this would be a very interesting and relevant topic to consider in a another work. This is especially true given the revised conclusions which we drew after unfolding, as these appear to differ from the conclusions drawn by other authors using othogonal polynomials, including those of 1312.5848v2.

R: I do not agree however, that theirs is just a result on "Chern-Simons matrix models". The model is of broader interest, even if that is perhaps not fully appreciated. One interesting aspect of the whole random matrix formulation of Chern-Simons theory, which has not been described elsewhere, is that somehow gives a sort of common ground between several approaches in quantum topology: It is well known that quantum topology can be equivalently approached in seemingly very different manners (and certainly, in many mathematical circles Chern-Simons theory is not precisely the preferred one). Notice the similarity or even equivalence of the methods and objects studied in this paper and the previous one (and some references therein) and the results by the knot theory group of H.R. Morton, which uses skein theory. See for example https://arxiv.org/pdf/math/0108011.pdf (especially Section 4). While the starting point is very different both approaches end up requiring evaluations of minors of structured matrices with Gaussian entries. This work handles this type of evaluations quite well, with new mathematical results.

The same minors appear in very different, and physically relevant, contexts, see for example the expression (93) in https://arxiv.org/pdf/2103.02545.pdf and the discussion around it, for another take on the interest in computing these minors in specific starting and ending configurations (the representations), precisely the kind of computation carried out here.

The Author's evaluation is also the evaluation of a Homfly polynomial of a torus link (as clearly explained by the authors) and, at least for torus knots, its homfly polynomial can be written in powers of a variable x=sqrt(q)-1/sqrt(q) and the corresponding coefficients have independent interest. In particular, for large N, the leading coefficient is p0 and it is a polynomial invariant itself, relevant in the study of periodic knots (studied by P. Traczyk and others). Taking also q->1 this object is a Wilson loop invariant in a 2d YM theory. This is speculative, but I wonder if, in their double scaling limit, and for the torus link invariant studied here, any of these considerations would apply ? If so, maybe that would be a further application of their analytical results.

The physical consequences of their detailed SFF analysis are perhaps unclear (to this reader at least) and I would say that, in spite of the obvious efforts by the authors, the paper is overall a bit difficult to read, no doubt due to its interdisciplinarity while also being quite technical in parts of it.

A: We thank the referee for the references, we were not aware of these. We aim to consider the suggestions made and see whether further applications can be found.

From the analysis described in more detail in our response to the first referee, our main physical conclusion would be that, after unfolding and for all $y$ except $y\approx 1$, the connected SFF of Chern-Simons matrix model exhibits Wigner-Dyson universality to very high precision, as it is very close to a linear ramp of unit slope. This is in spite of the fact that this matrix model was introduced and studied in the RMT literature to describe the intermediate eigenvalue statistics of strongly disordered electrons. Taking $q \to 1^-$, as in the `t Hooft limit, should correspond to the greatest deviation from the CUE ($q\to 0)$. The fact that Wigner-Dyson universality appears to be recovered in the `t Hooft limit is therefore quite unexpected.

R: Since one of the difficulties with this paper is that, content-wise, it may be quite hard for the intended audience, together with the difficulties in advancing the somewhat elusive physics of these intermediate ensembles (beyond their exact appearances, above mentioned, and in Chern-Simons theory), that suggests that perhaps it would be a better fit in a mathematical physics journal, where it could be a very good contribution.

---

## Round 2 · Referee Report · Anonymous · 2022-6-10

Strengths
After a more careful analysis the authors reached the opposite conclusion that the eigenvalue stats are Wigner-Dyson after
the eigenvalues have been rescaled.
Weaknesses
Figs. 1, 2 and 4 are for spectra that have not been unfolded. This
is misleading and suggests that the spectral statistics are not
Wigner Dyson.
The unfolding was done the poor man's way by just rescaling.
The level density is close to a semicircle, and it is simple
to do an exact numerical unfolding.
Report
Figures 1, 2 and 4 need to be replaced by figures with the
connected spectral form factor for unfolded eigenvalues.
The level density is close to a semi-circle, and it is easy to
unfold the eigenvalues by fitting a semicircle times a
low order even polynomial to the spectral density. Although
N=10 or 20 is not large, I think that the result will already
be close to the Wigner-Dyson result. Numerically, it should
be no problem to go to N=100 and in that case the agreement
should be very good. The authors could keep the original
figures, but then should add the figures with unfolded spectra
next to it.
With regards to integrable systems there are many that do not
have Poisson statistics. For example harmonic oscillators with
commensurate frequencies, or many-body systems with energies given by the sum of single particle energies.
Requested changes
Already mentioned that above

---

## Round 2 · Author Response

This paper has 30 pages (in its arxiv version) and 6 figures

---

## Round 2 · List of Changes

The changes listed here are described in more detail in our responses to the referees.
Changes:
1. Added a treatment of unfolding and revised our conclusions, changed title, abstract, introduction, and conclusion to align with revised conclusions
2. Removed comparison with linear fit to connected SFF, including figure
3. Changed commas to decimal points
4. Added figures on level density and unfolded SFF
5. Added references
6. Corrected typos

You are currently on this page

---

## Round 3 · Author Response

We have added the numerically unfolded spectral form factor (SFF) for N=10 and N=20, as per the referee's suggestion. The 'poor man's' unfolded SFF in the 't Hooft limit is given in figure 7, which we have clearly indicated in the text. We intend to defer a more extensive numerical analysis to a future publication.

---

## Round 3 · List of Changes

1. Added a treatment (including plots) of numerically unfolded SFF's for N=10 and N=20.
  2. Corrected typos

---

## Editorial Decision

published